# Few-shot Algorithms for Consistent Neural Decoding (FALCON) Benchmark

**Brianna M. Karpowicz**[1,2*]   **Joel Ye**[3*]   **Chaofei Fan**[4]   **Pablo Tostado-Marcos**[5]
**Fabio Rizzoglio**[6]   **Clay Washington**[1,2]   **Thiago Scodeler**[7]   **Diogo de Lucena**[7]
**Samuel R. Nason-Tomaszewski**[1,2]   **Matthew J. Mender**[8]   **Xuan Ma**[6]
**Ezequiel Matias Arneodo**[5,9]   **Leigh R. Hochberg**[10−12]   **Cynthia A. Chestek**[8]
**Jaimie M. Henderson**[4]   **Timothy Q. Gentner**[5]   **Vikash Gilja**[5]   **Lee E. Miller**[6]
**Adam G. Rouse**[13]   **Robert A. Gaunt**[14]   **Jennifer L. Collinger**[3,14]   **Chethan Pandarinath**[1,2†]

[1]Emory University   [2]Georgia Tech   [3]Carnegie Mellon University   [4]Stanford University
[5] University of California San Diego   [6] Northwestern University   [7] Agency Enterprise Studios
[8] University of Michigan   [9] Instituto de Física La Plata   [10] Harvard Medical School
[11] Department of Veterans Affairs   [12] Brown University
[13] University of Kansas Medical Center   [14] University of Pittsburgh

## Abstract

Intracortical brain-computer interfaces (iBCIs) can restore movement and communication abilities to individuals with paralysis by decoding their intended behavior from neural activity recorded with an implanted device. While this activity yields high-performance decoding over short timescales, neural data are often nonstationary, which can lead to decoder failure if not accounted for. To maintain performance, users must frequently recalibrate decoders, which requires the arduous collection of new neural and behavioral data. Aiming to reduce this burden, several approaches have been developed that either limit recalibration data requirements (few-shot approaches) or eliminate explicit recalibration entirely (zero-shot approaches). However, progress is limited by a lack of standardized datasets and comparison metrics, causing methods to be compared in an ad hoc manner. Here we introduce the FALCON benchmark suite (Few-shot Algorithms for COnsistent Neural decoding) to standardize evaluation of iBCI robustness. FALCON curates five datasets of neural and behavioral data that span movement and communication tasks to focus on behaviors of interest to modern-day iBCIs. Each dataset includes calibration data, optional few-shot recalibration data, and private evaluation data. We implement a flexible evaluation platform which only requires user-submitted code to return behavioral predictions on unseen data. We also seed the benchmark by applying baseline methods spanning several classes of possible approaches. FALCON aims to provide rigorous selection criteria for robust iBCI decoders, easing their translation to real-world devices. https://snel-repo.github.io/falcon/

## 1 Introduction

Brain-computer interfaces (BCIs) provide a path to restore movement and communication in individuals with paralysis by decoding neuronal population activity to uncover the user's intention. BCIs have recently achieved many promising demonstrations, including high degree of freedom robot arm control [1, 2], computer use and communication [3–8], and speech decoding [9–13]. A specific class of BCIs known as intracortical BCIs (iBCIs) have enabled many of these impressive technological feats. However, many of these demonstrations have required decoders to be recalibrated daily or

---

*Equal contributions.   Correspondence to †chethan@gatech.edu

38th Conference on Neural Information Processing Systems (NeurIPS 2024) Track on Datasets and Benchmarks.

even more frequently, interrupting device use and burdening the user. Real-world iBCI deployment will require maintaining high performance over long time periods with minimal recalibration. The challenge here stems from nonstationarities in the neural data that are caused by many factors acting at multiple timescales, such as shifts in the position of the electrode relative to surrounding tissue, changes in tissue properties in response to electrode implantation, electrode malfunction, or neural plasticity [14, 15]. These nonstationarities result in a changing relationship between neural data and behavior, necessitating frequent decoder recalibration to maintain high performance.

Fortunately, despite these nonstationarities, there are many potential ways to leverage structure in neural or behavioral data to help reduce the burden of recalibration [16]. For example, while the spiking activity recorded on an individual iBCI electrode can change over timescales of hours, neural population activity contains low dimensional structure (manifolds) that shows a consistent relationship with behavior over months to years [17–19]. Realignment methods that exploit these conserved manifolds [20–26]can restore decoding without explicit calibration periods. Alternatively, rather than focusing on structure intrinsic to the neural data, another set of approaches attempts to achieve robustness by continually recalibrating decoders using the retrospective analysis of data collected during the subject's normal use of the iBCI [27–29]. A third strategy has centered on supervised deep network training using many sessions to yield decoders that are robust to session-to-session variability [30, 31], which may be extended further to potentially yield universal iBCI decoders that generalize to new subjects or tasks [32–34]. These diverse and potentially complementary efforts converge around a single problem statement: the real-world iBCI decoding challenge is to maintain high performance on distinct, but related, data distributions, with minimal data from the new setting.

While these diverse approaches have thus far used their own ad-hoc evaluation, standardization could enable rigorous comparison to assess real-world potential and highlight advances upon which future efforts can build. We propose the FALCON benchmark, **F**ew-Shot **Al**gorithms for **Con**sistent **N**eural Decoding, as a common evaluation for stable, long-term decoding performance. FALCON releases 5 multi-session datasets that span movement and communication tasks relevant to iBCIs: human and monkey reach and grasp behavior (H1, M1), monkey finger movement (M2), human handwriting (H2), and birdsong (B1). These datasets are divided into held-in and held-out sessions. To evaluate how well few-shot decoders advance iBCI robustness given real-world data constraints, only a small amount of supervised data is released from held-out sessions. Approaches for the more challenging settings of only using neural data on new sessions (unsupervised), or no data from new sessions (zero-shot), can also be evaluated using the same data splits. This report describes the design of the benchmark, its datasets, and the performance of baseline models. By introducing FALCON, we aim to establish standardized evaluation practices for robust iBCI decoding approaches that can provide researchers with metrics to select methods for in-device use.

## 1.1 Related work

**Benchmarks of BCI Decoding.** FALCON evaluates iBCI decoding, or the prediction of intention from neural activity. To date, benchmarks of decoding have been uncommon compared to other fields using machine learning. The early BCI competition series [35] and more recent additions of the International BCI Competition [36] and MOABB [37] evaluated decoding in offline (i.e., pre-recorded) noninvasive neural datasets in multiple subjects and highlighted several of the challenges faced in iBCI datasets (multi-session transfer, removal of repeated data structure). More recently, the Brain2Text decoding benchmark [38] evaluates speech decoding in human iBCIs. However, absent a strong benchmarking culture, models across intracortical and noninvasive neural recording modalities [32–34, 39? –42] are still often evaluated on different public or private datasets. This lack of standardization makes comparison across works difficult due to subjectivity in preprocessing, metric choice, and evaluation design.

**Benchmarks on Neural Data.** Benchmarks for neural data analysis are related but differently motivated than decoding benchmarks. BrainScore [43] evaluates the ability of models to predict brain data when trained on non-neural data tasks. The Sensorium [44] and Algonauts [45] challenges evaluate encoding models that predict brain activity of mouse visual cortex and human fMRI, respectively, given visual stimuli. The Neural Latents Benchmark (NLB) [46] evaluates latent variable models on spiking activity from different brain areas of monkeys. While the NLB has a decoding metric, this metric is computed with ridge regression on inferred latent variables and is

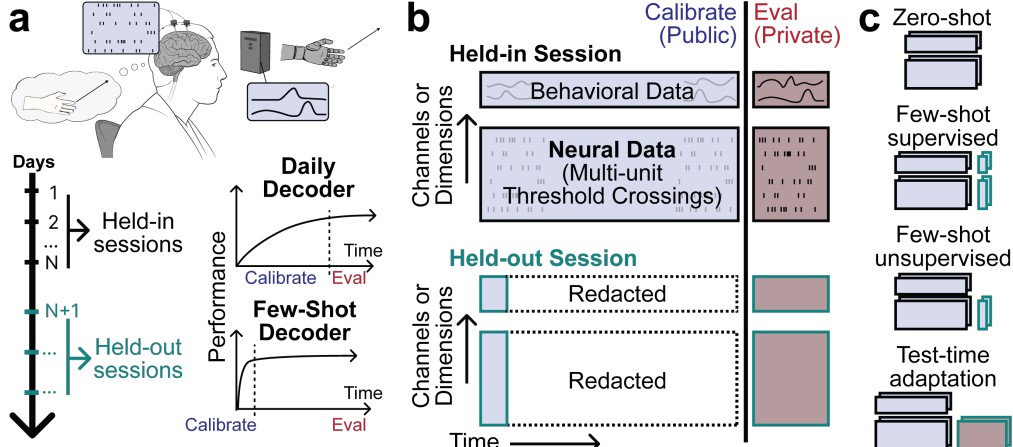

**Figure 1. FALCON Evaluation Design. (a)** Top: BCI decoders are prepared by collecting calibration data where a user attempts to perform a cued behavior. This process yields paired examples of behavioral outputs and associated neural data. Bottom: Current practice requires new calibration data to train new decoders. High decoding performance may require substantial data, motivating methods for few-shot decoding. FALCON provides full "held-in" sessions and evaluates few-shot decoding on "held-out" sessions. **(b)** Each session in a FALCON dataset contains multiunit threshold crossings and behavioral data (which are discrete sentences in H2 and continuous covariates otherwise). Evaluation data is withheld from all sessions. All remaining data is released publicly for held-in sessions. Only a small fraction of data is released for held-out sessions. **(c)** FALCON's design enables comparison of different approaches for consistent decoding. Zero-shot methods use no data from held-out sessions, few-shot methods use the calibration splits from held-out sessions, and test-time adaptive methods can implement behavior-free, unsupervised decoder updates during evaluation.

not treated as a primary endpoint. FALCON directly evaluates decoding, allows more flexibility in decoder architecture, and more closely aligns with the goal of evaluating the quality of iBCI decoders.

## 2 Benchmark evaluation pipeline and metrics

### 2.1 Evaluation strategy and pipeline

FALCON evaluates behavioral decoding from iBCI neural activity in five datasets. Each dataset comprises multiple sessions of data divided into two contiguous splits: held-in and held-out (**Fig. 1a**). As in standard decoder calibration, held-in sessions provide sufficient data to train a high-performing decoder; held-out sessions are prepared for evaluating few-shot decoder performance and therefore include insufficient data to prepare decoders from scratch ( **Fig. 1a,b**). All datasets provide multi-unit threshold crossings (detected voltage deflections caused by nearby neuron action potentials) recorded from intracortical electrodes and behavioral data (specified per task). An evaluation split of the same length is withheld from both held-in and held-out. All remaining data is released for held-in sessions while a small fraction of data is released for held-out sessions. Note that the held-in split provides a standard-data regime iBCI decoding benchmark, but FALCON focuses on few-shot decoding performance in the held-out split.

Submitted decoders are executables that implement an iBCI prediction interface. The evaluation server (EvalAI [47]) requires causal, open-loop predictions to be made on streaming neural data, timestep-by-timestep. The communication datasets make predictions on coarser timescales (per sentence for H2, per song motif for B1). These formats mimic current iBCI use for their respective tasks. Lack of trial structure in movement tasks is an important training time consideration; decoders trained on trialized data can degrade significantly when evaluated continuously (Section A.5.1). We note that an important limitation of FALCON is that evaluation may be susceptible to promoting models that exploit trial structure implicit in the datasets, even if this does not benefit iBCI control [48].

## 2.2 Supported approaches and benchmark scope

FALCON allows methods with varying data assumptions to be evaluated in a common setting (**Fig. 1c**). Decreased data use provides greater reduction of user burden, but can be more challenging.

**Zero-shot** methods directly predict behavior on new sessions with fixed model parameters. This typically requires using deep networks that train on many sessions (e.g. months) of data. Such methods have enabled high performance cursor control on new days for months into the future [30, 31]. Recent efforts exploring subject generalization [49, 50, 34] and neural data foundation models [32, 33] may alleviate the burden of large scale data collection on individual users. However, as current multi-session zero-shot methods impose large data collection burdens on the user and may still degrade after long-term use, there is a practical need to explore adaptive methods as well.

**Few-shot supervised** methods assume the collection of limited calibration data for every session of use. For people with paralysis, a high-performance iBCI that requires a short calibration procedure before use may still confer a large advantage over other assistive technologies. Current supervised deep networks typically adapt to short calibration blocks through fine-tuning of a pretrained model [32–34]. Few-shot supervision is the least strict setting that can be evaluated with FALCON that still reduces user burden, for which the highest performance is expected.

**Few-shot unsupervised** methods remove the need for behavioral data on new days, skipping explicit calibration periods by allowing recalibration procedures to be performed using only neural data from normal iBCI use. Due to their lack of reliance on behavioral data, unsupervised approaches are not subject to problems that may arise from behavioral labels, which may be difficult to obtain during iBCI use when guessing a user's intent post-hoc can be unreliable. Unsupervised methods typically assume that the neural activity has an underlying manifold which maintains a stable relationship to behavior over long periods of time [20–25]. However, the specific context of iBCI use, such as strategy or posture, may lead to a change in the manifold-to-behavior mapping and violate this assumption. FALCON's datasets are drawn from consistent behavior across days, though due to behavioral complexity, not all behavioral conditions will be sampled in the few-shot calibration data.

**Test-time adaptation** generally leverages behavioral priors to provide model labels on unlabeled data. Currently proposed test-time adaptation methods avoid the collection of any calibration data on test days. Instead, these methods use neural data and inferred behavioral labels collected during normal iBCI use to perform "semi-supervised" decoder recalibration. However, these methods have only been demonstrated for two-dimensional cursor use and language communication [27–29], suggesting open challenges for broad behavioral domains, such as in FALCON's movement datasets.

## 2.3 Metrics

Each dataset uses a standard decoding metric. The movement tasks (M1, M2, H1) require predictions of multi-dimensional motor covariates, such as muscle activity. For these tasks, accuracy is reported using the coefficient of determination ($R^2$), computed as a variance-weighted average across the $R^2$ of individual motor covariates. $R^2$ is useful for interpreting low-dimensional predictions as a constant mean prediction achieves an $R^2$ of 0 and max $R^2$ is 1. The handwriting task (H2) requires prediction of English characters from a corpus of common sentences; we use word error rate (WER) as a metric, computed as the edit distance between the predicted and expected sequence divided by the length of the intended sequence. Birdsong decoding (B1) reports performance as mean squared error (MSE) on the predicted spectrogram; MSE is preferable for evaluating spectrogram predictions as the predictions are much higher-dimensional than in movement tasks. Metrics are computed per session, and across-session mean and standard deviation are reported on EvalAI. Mean and standard deviation are computed separately for held-in and held-out splits.

## 3 Datasets

FALCON aims to provide a comprehensive evaluation of few-shot decoding across contemporary iBCI applications. FALCON datasets span two primary groups of tasks: movement and communication (**Fig. 2**). FALCON's movement datasets have either kinematic or muscle outputs, and the communication datasets have either text or vocal outputs. Because most human iBCI study participants have limited independent movement, human behavioral data are those that the researcher asked the participant to attempt or imagine, while animal behavioral data are recorded from physical

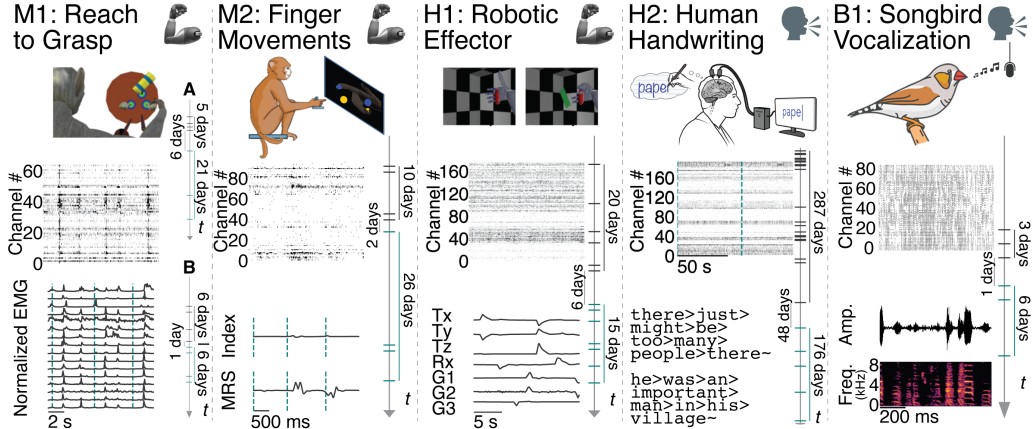

**Figure 2. FALCON datasets span iBCI use cases.** For each column, *top*: task schematic; *middle*: neural activity for all channels over time; *bottom*: example behavioral outputs. Each panel includes a vertical timeline denoting held-in (gray) and held-out (teal) sessions in the dataset. Ticks mark individual sessions, colored vertical bars indicate time elapsed within or between splits. **Tasks:** Movement datasets (mechanical arm) include: monkey reach-to-grasp (M1, 2 monkeys (A/B), 64/96 channels neural data, 16 channels muscle activity), monkey finger movements (M2, 96 channels neural data, 2-dimensional finger movements), and human robotic effector (H1, 172 channels neural data, 7-dimensional hand and arm velocity outputs). Communication tasks (speaking head) include human handwriting (H2, 192 channels neural data) and songbird vocalization (B1, 85 channels neural data).

actions. All datasets contain electrophysiological voltage recordings collected from intracortical microelectrodes. We extract threshold crossings from the recorded voltages to yield spiking activity, as is standard practice for iBCIs [51]. Detailed descriptions of each dataset and their locations can be found in Section A.3.

While the ultimate goal of some iBCI research is applications in humans, we provide animal datasets because animal models are essential to develop iBCI applications and for basic scientific discovery [52]. Using both animal and human data also improves the likelihood of finding models with broad effectiveness, as levels of instability are likely to vary across subjects and species [22, 23, 53].

**M1: Monkey reach and grasp.** The M1 dataset consists of recordings using Floating Microelectrode Arrays (Microprobes), implanted in the precentral gyrus while two monkeys (M1-A and M1-B) reached to, grasped, and manipulated an object in a specific location (4 possible objects, 8 possible locations) [54–57]. Intramuscular electromyography (EMG) was recorded from 16 muscles in the right hand and upper extremity. The large number of object/location combinations leads to a wide variety of muscle activations. Unlike higher-level behavioral variables (such as robotic arm endpoint velocities), EMG is a directly measurable output of the motor nervous system, and thus provides a signal that should have a close correspondence to neural activity on a moment-by-moment basis. EMG is also directly relevant to iBCIs that combine with functional electrical stimulation to control paralyzed limbs [58, 59]. Monkey EMG is interesting to iBCI research as human iBCI users with paralysis have limited muscle control and likely lack the ability to produce EMG decoding targets; recent works have proposed cross-species transfer to exploit monkey EMG data for iBCI applications [49].

**M2: Monkey finger movements.** The M2 dataset consists of Utah array recordings from the precentral gyrus while a monkey made finger movements to control a virtual hand to acquire cued target positions [60, 61]. Finger actuation ranged from full extension to full flexion with cued movements focusing on the index finger and/or the middle-ring-small (MRS) finger group. The goal of including M2 in the FALCON benchmark is to develop methods that accurately predict individuated finger movements over time. Finger control is a critical aspect of dexterous hand function and is a key target for iBCI control that aims to restore upper limb and hand function to individuals. Recent work has shown that the encoding of finger behaviors in motor cortex may be compositional [60, 62]; yet, the implications of this finding on iBCI control and decoding stability are unclear.

**H1: Human robotic effector.** The H1 dataset contains Utah array recordings from the hand and arm motor cortex of a human iBCI participant, collected in a long-term clinical study on iBCIs for sensorimotor control. The participant was cued to attempt to reach and grasp with their right

hand. This data was used to calibrate an iBCI for control of a robotic arm in a 7 degree-of-freedom task [1, 63, 64]. These data are open loop, meaning that the participant attempted cued movements but was not directly controlling the output and could not correct errors in real-time. H1 contains a breadth of combinations of robotic arm command variables (3D limb kinematics, 1D rotation, 3D grasp shape) that are often decoding targets for iBCIs. High-dimensional control is particularly burdensome to calibrate, as the large number of possible endpoints demands calibration procedures that are often several minutes long [63]. Developing methods to improve the efficiency of calibration to novel sessions would advance the practical viability of using iBCIs for high-dimensional control.

**H2: Human handwriting.** The H2 dataset contains neural activity recorded using Utah arrays placed in the "hand knob" area of the dorsal motor cortex of a human iBCI participant, collected as part of the BrainGate2 Clinical Trial. The participant was asked to copy a sentence by attempting to write each letter individually [5]. The H2 dataset falls in the domain of brain-to-text BCIs, which aim to restore communication capabilities. Decoders for this task need to accurately predict the intended character as well as determine when that character was intended to be written, as the task is fully self-paced. This task is therefore not amenable to traditional linear decoders and will require more sophisticated approaches, most canonically RNN decoders with a Connectionist Temporal Classification loss [5, 10, 11, 29]. Additionally, due to the goal of predicting words or sentences, communication iBCIs often use large language models to further refine predictions or build stable decoders [10, 29].

**B1: Songbird vocalization.** The B1 dataset features neural recordings from a zebra finch songbird using Neuropixels 1.0 probes [65] implanted in the motor brain region robust nucleus of the arcopallium (RA). Alongside neural activity, this dataset includes simultaneous free-behavior audio recordings during awake-singing. Songbird neuroanatomy and vocal behavior have direct parallels to human speech [66], thereby offering a valuable model for exploring neurally-driven speech synthesis applications. The B1 dataset presents a unique challenge for stable decoding approaches. Neuropixels probes may exhibit nonstationarities that vary significantly in type and timescale compared to traditional microelectrode arrays. Given vocal ground-truth, decoders designed for B1 aim to synthesize high-fidelity continuous amplitude waveforms or spectrogram representations of vocal output. This strategy may better preserve the prosodic elements in reconstructed vocalizations, a significant challenge inherent to current brain-to-text approaches. By developing stable birdsong decoding strategies, we aim to establish baseline methods that can be adapted to human brain-to-speech iBCIs.

## 4 Results

We seed FALCON with representative approaches to provide an initial characterization of the stability challenge. Implementation details on all baselines are provided in Section A.4. For all datasets, we provide standard decoders applied to held-out sessions in two ways: (1) trained in a many-shot manner using redacted data ("oracle" decoders) approximately upper-bound performance and (2) applied zero-shot ("static" decoders) to lower-bound performance. For motor datasets (M1, M2, H1), we fit a Wiener Filter (WF; ridge regression with history) on inferred neural firing rates derived from an exponential spike smoothing kernel (see Section A.4.1). A single-session WF is a simple but effective baseline for offline decoding from high quality spiking activity and is a representative default method for closed loop control. We also train a single-session recurrent neural network (RNN) and a multi-session Neural Data Transformer (NDT2 Multi [32]) to establish the performance of higher capacity nonlinear models. For the human handwriting dataset H2, we provide an RNN trained on multiple sessions to predict English letters from neural activity, and a second RNN that additionally uses language models (LMs) as priors to correct RNN outputs and improve accuracy. On B1, we apply the EnSongdec decoder [67] which predicts song embeddings from spiking data using a feedforward network and synthesizes them into continuous birdsong using a pretrained EnCodec model [68].

We also sample state-of-the-art methods for robust decoding to demonstrate how existing approaches perform on FALCON datasets. For movement datasets, we provide two deep-network-based unsupervised few-shot alignment approaches, Nonlinear Manifold Alignment with Dynamics (NoMAD) [22] and Cycle-consistent Generative Adversarial Network (CycleGAN) [23]. Similar to neural latent variable models [46], NoMAD and CycleGAN use an RNN and an MLP, respectively, to infer neural firing rates through a Poisson firing rate model. These methods then apply distributional alignment to match inferred neural firing distributions on held-out sessions to those of held-in sessions. Different single-session models are trained to provide the different held-in scores. Additionally, we train NDT2

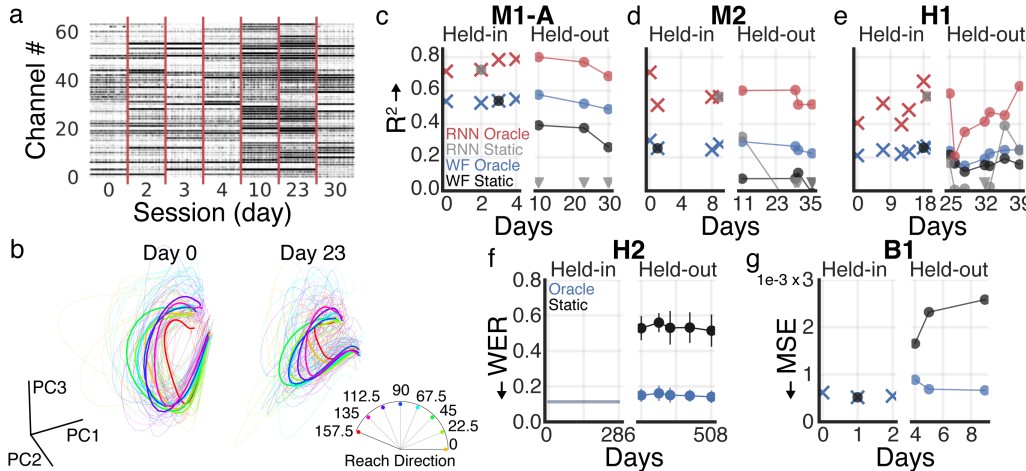

**Figure 3. Static decoders exhibit decoding instabilities on FALCON datasets.** (a) Raster plot showing 1 minute of data for each `M1-A` session, separated by red vertical lines. (b) Neural trajectories from PCA fit on `M1-A` Day 0 smoothed spiking activity and applied to Day 0 and Day 23 smoothed spiking activity. Colored by reach direction. Thick lines show the average of all reaches in a given direction and thin lines indicate single reaches. (c-e) $R^2$ of oracle decoders (RNN red, WF blue) and static decoders (RNN gray, WF black) for held-in and held-out splits for `M1-A`, `M2`, and `H1`. Higher values indicate more accurate performance. Downward triangles indicate points with negative $R^2$ otherwise not visible on these axes. Selected static decoders are annotated with a gray or black circle. (f) Word error rate (WER) of oracle (blue) and static (black) decoders for `H2`. Lower values indicate more accurate performance. Rather than training one model per held-in session, the held-in decoder is trained using all held-in sessions (performance denoted by horizontal line). Held-out dataset performance reported as mean $\pm$ standard deviation across 5 random seeds. (g) Mean squared error (MSE) of oracle (blue) and static (black) EnSongdec models for `B1` held-in and held-out splits. Lower values indicate more accurate performance. Static decoder chosen from held-in datasets indicated with black circle.

Multi models that only use calibration data. The `H2` stability baseline is a test-time adaptive method that uses the LM-corrected outputs as pseudo-labels to iteratively recalibrate the RNN (Continual Online Recalibration with Pseudo-labels; CORP [29]). As vocalization decoding has seen limited development of specific decoder stabilization approaches, we pose `B1` as an open question and solicit potential solutions from FALCON submissions.

## 4.1 FALCON datasets exhibit unstable decoding performance across sessions.

We first show that FALCON datasets exhibit qualitative nonstationarities, reflecting the challenges faced in iBCI use. **Figure 3a** shows neural spiking activity from all sessions in the `M1` dataset. It is clear that neural firing exhibits different properties across sessions. We also visualize this data in 3 dimensions using principal components analysis (PCA) (**Fig. 3b**). Under a common projection, we plot average time courses for different reach directions. Directions are clearly separable in both sessions (supporting decoding), but the required decoding map changes between sessions.

Next, we quantify that each session's neural activity can provide good decoding of behavior. We train oracle decoders for each session, which use all non-evaluation data. Specifically, held-in oracle decoders use the data from the calibration split, and held-out oracle decoders use both the calibration split and the redacted data. All oracle decoders are evaluated on respective session evaluation splits. For movement datasets, oracle decoders consist of Wiener Filters with cross-validated history (WFs) or single-session RNN models (**Fig. 3c-e**, blue/red). For `H2`, the oracle decoder is an RNN trained to predict letters from neural data, trained jointly on all held-in calibration splits and incrementally with each session's held-out calibration and redacted splits (**Fig. 3f**, blue). For `B1`, we apply an EnSongdec model [67], which uses neural data to predict song embeddings before reconstructing song spectrograms (**3g**, blue). For all datasets, variability in oracle decoding performance is nontrivial but small, implying that performance drops from transferring decoders to new sessions are not due to degraded neural data or a lack of correspondence between neural and behavioral data.

Finally, we quantify decoding instabilities in each dataset with zero-shot static decoders. The specific static decoder was chosen from the held-in session oracle decoders as the highest performing on

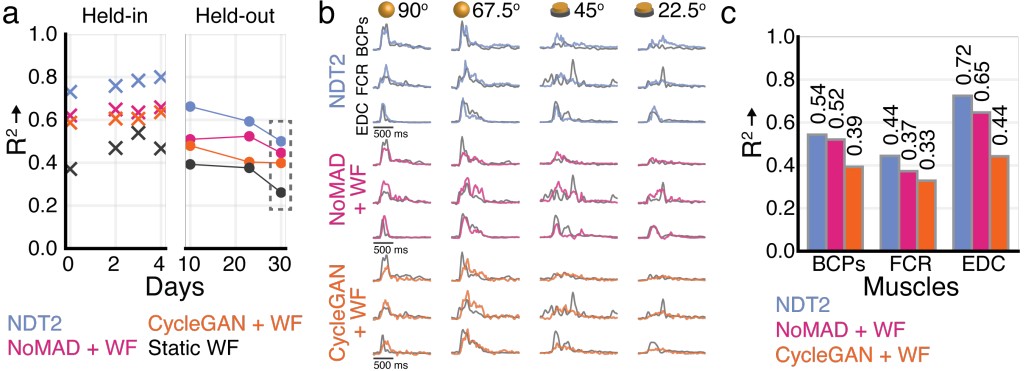

**Figure 4. Baseline model predictions on `M1-A` dataset.** (a) Performance ($R^2$) of each baseline model on individual held-in and held-out `M1` datasets. Box indicates the dataset that will be elaborated on in later panels. (b) Example decoded EMG traces for each baseline approach. Three of the sixteen total muscles shown: biceps (BCPs), flexor carpi radialis (FCR), and extensor digitorum (EDC). Each column is an example trial for one object (sphere or button) and location (angle) pair. Gray traces are the measured EMG for that muscle and experimental condition, and colored traces are the EMG predicted by each decoder-stabilization method on Day 30. (c) $R^2$ values computed for three individual muscles. Together with the remaining muscles, these values comprise the variance-weighted $R^2$ presented in panel (a).

the evaluation split of the other held-in sessions, to approximate good generalization to held-out sessions. We apply the decoder unmodified to the held-out datasets, simulating an iBCI decoder's naive (zero-shot) performance on a new session without recalibration. All datasets showed marked decoding instability (**Fig. 3c-g**, black/gray), with drops in WF performance up to 0.28 $R^2$ (`M1`), 0.27 $R^2$ (`M2`), and 0.14 $R^2$ (`H1`). RNN decoders exhibit more instability on FALCON movement datasets, with drops up to 1.94 $R^2$ (`M1`), 0.82 $R^2$ (`M2`) and 0.78 $R^2$ (`H1`). Communication datasets demonstrate similar trends – error increases as much as 0.40 WER (`H2`) and 9.03e-4 MSE (`B1`).

### 4.2 FALCON baselines demonstrate the difficulty of improving `M1` decoder stability.

We next compare current few-shot approaches applied to `M1-A` in detail. On the held-in datasets, NDT2 yields the highest performance, followed by NoMAD + WF, and CycleGAN + WF (**Fig. 4a**). From held-in to held-out sessions, NDT2 dropped by at most 0.30 $R^2$, NoMAD by at most 0.21 $R^2$, and CycleGAN by at most 0.24 $R^2$. Compared to the static WF (drop $\leq$ 0.28 $R^2$), the baseline approaches show at most a marginal improvement, indicating that stability challenges still affect all approaches applied to `M1`. Model ranking and relative performance are largely preserved across sessions, implying that averaging $R^2$ across sessions summarizes performance without obscuring gains on specific sessions.

For Day 30, we also present decoded predictions for three key muscles - the biceps (BCPs), flexor carpi radialis (FCR), and extensor digitorum (EDC) - for each baseline approach. In **Fig. 4b**, each column is an individual reach for one of the location-object pairs available in the evaluation split on Day 30. Comparing the predicted EMG traces to the measured EMG traces provides context for interpreting the $R^2$ numbers and understanding which features of the EMG (the baseline, the high frequency features, the magnitude) are predicted well by each method. For example, NDT2 captures more high frequency changes in the muscle activity than other methods, potentially due to its nonlinear decoding. In **Fig. 4c**, we show $R^2$ values for example muscles individually. Per-muscle performance preserves the method ranking shown in **Fig. 4a**, providing further confidence that the variance-weighted $R^2$ over output dimensions is sound.

### 4.3 FALCON baseline performance drops from held-in to held-out datasets.

Baseline results on all datasets are shown in **Table 1**. FALCON quantifies notable performance gaps across methods. For example, within oracle decoders, which are by definition trained using the same data, increased model complexity can substantially improve decoding (`M2`: 0.27 vs 0.77 $R^2$ WF/NDT2 Multi; `H2`: 0.11 vs 0.02 WER RNN Multi/+ LM). Moreover, it is unsurprising that using

**Table 1. FALCON baselines**. Metric means and standard deviations over sessions, computed for held-in data and held-out data separately. Standard deviations only shown for the held-out split, for clarity. *Metrics*: $R^2$ for movement tasks, word error rate (WER) for H2, mean squared error (MSE) for B1. *OR*: oracle models trained with unreleased data on held-out split. *ZS*: Zero-shot/static. *FSU*: Few-shot unsupervised. *FSS*: Few-shot supervised. *TTA*: Test-time adaptive. *Multi*: denotes training with multiple held-in datasets; otherwise models use a single held-in dataset.

**Movement (Held-Out $R^2$ / Held-In $R^2$ ↑)**

| | Class | M1-A | M2 | H1 |
|---|---|---|---|---|
| Wiener Filter (WF) | OR | $0.53_{\pm0.04}/0.54$ | $0.26_{\pm0.03}/0.27$ | $0.21_{\pm0.04}/0.24$ |
| RNN | OR | $0.75_{\pm0.05}/0.75$ | $0.56_{\pm0.04}/0.59$ | $0.44_{\pm0.13}/0.51$ |
| NDT2 Multi | OR | $0.78_{\pm0.04}/0.77$ | $0.58_{\pm0.04}/0.62$ | $0.63_{\pm0.08}/0.68$ |
| WF | ZS | $0.34_{\pm0.06}/0.46$ | $0.06_{\pm0.04}/0.15$ | $0.16_{\pm0.03}/0.20$ |
| RNN | ZS | $-.60_{\pm0.45}/0.52$ | $-0.07_{\pm0.23}/0.20$ | $0.09_{\pm0.18}/0.31$ |
| CycleGAN + WF [23] | FSU | $0.43_{\pm0.04}/0.61$ | $0.22_{\pm0.06}/0.32$ | $0.12_{\pm0.06}/0.15$ |
| NoMAD + WF [22] | FSU | $0.49_{\pm0.03}/0.64$ | $0.20_{\pm0.10}/0.35$ | $0.13_{\pm0.10}/0.21$ |
| NDT2 Multi [32] | FSS | $0.59_{\pm0.07}/0.77$ | $0.43_{\pm0.08}/0.63$ | $0.52_{\pm0.04}/0.62$ |

**Communication (Held-Out Error / Held-In Error ↓)**

| | Class | H2 (WER) | | Class | B1 (MSE $\times10^{-4}$) |
|---|---|---|---|---|---|
| RNN Multi | OR | $0.15_{\pm0.01}/0.11$ | EnSongdec [67] | OR | $7.47_{\pm0.99}/5.61$ |
| RNN Multi + LM | OR | $0.03_{\pm0.00}/0.02$ | | | |
| RNN Multi | ZS | $0.53_{\pm0.02}/0.11$ | EnSongdec | ZS | $21.8_{\pm3.91}/5.18$ |
| RNN Multi + LM | ZS | $0.37_{\pm0.01}/0.02$ | | | |
| CORP [29] | TTA | $0.11_{\pm0.01}/0.02$ | | | |

more data will provide large performance gains (ZS to FSU to FSS in movement datasets, ZS to TTA in H2). FALCON encourages the submission of novel approaches in each class of data use.

Given FALCON's flexibility to accommodate many classes of approaches, a method's held-in score may be used to contextualize its own held-out score. Oracle decoders establish the approximate variability in performance between held-in and held-out splits, which appears relatively small (e.g., max difference = 0.04 $R^2$ for NDT2 Multi on H1, 0.04 WER for RNN Multi on H2). Yet, all methods show sizable gaps between held-in and held-out scores, far exceeding the expected variability. In absolute terms, all decoders perform well on held-in M1 ($R^2$ = 0.46-0.78), but performance drops by 0.12-0.18 $R^2$ on the held-out split (and the RNN has an extreme failure). M2 and H1, which show lower overall decoding performance, maintain that held-out scores are only a fraction of the potential performance indicated by held-in scores. This is also true for H2, where CORP provides a great advance over zero-shot methods but yields error on the held-out datasets that is nearly 4x higher than that of oracle decoders on average (0.03 vs 0.11 WER). These results indicate that room for improvement remains in the few-shot challenge on FALCON datasets.

## 5   Discussion

FALCON extends previous efforts to benchmark models of neural data by presenting a standardized evaluation procedure for algorithms that improve decoder robustness in iBCI applications. We release datasets from 3 movement and 2 communication tasks, spanning monkeys, songbirds, and human participants. FALCON is designed to be inclusive of many classes of approaches; we demonstrate standardized comparison of 5 different approaches for movement datasets, 3 different approaches for H2, and 1 approach for B1, each with varying complexity and data-use strategies. These initial models far under-sample the wide design space of methods; we believe further submissions to FALCON will help clarify the value of different training data and priors. We hope that FALCON will encourage new approaches to be developed and adopted for real-world iBCI devices.

We expect that FALCON will enable machine learning researchers to apply cutting-edge approaches to a neuroengineering problem. To this end, we impose minimal restrictions on training strategies: we allow zero-shot, few-shot, or test-time adaptation and provide generous compute for model inference.

While the provided baselines train with only the calibration data, FALCON is compatible with foundation models and can be used to assess their efficacy for improving iBCI robustness.

**Extensions and limitations**    FALCON's datasets, except for B1, contain constrained behavior with trial structure, derived from repeated cues to start and stop stereotyped behavior. Model memorization of trial structure can impede closed loop control and has been a major hurdle for adopting deep networks across iBCI settings [48, 69, 70]. Corroborating this narrative, NDT2 models trained on trialized data degraded in FALCON's continuous evaluation (Section A.5.1). To penalize sensitivity to trial structure, FALCON does not provide trial labels in movement decoding tasks. However, FALCON datasets are still inherently structured. Providing datasets with more naturalistic behavior is technically challenging, particularly in humans without intact motor abilities for whom intended behavior must be communicated post-hoc. Nonetheless, future extensions may endeavor to evaluate more free and diverse behaviors, bringing evaluation closer to real-world iBCI use. To more easily aggregate a large number of behaviors, advances in cross-subject or cross-task generalization [49, 32, 50, 71] motivate analogous few-shot benchmarks where users are given restricted data for a new subject or behavior.

An important consideration in interpreting FALCON is that it evaluates open loop prediction, not closed loop iBCI control. Closed loop control introduces shifts in neural data due to sensory feedback [72] and consequent user compensation. Users can correct for certain classes of decoder error, implying that worse decoder predictions may not yield poor control [73, 74]. The popular robotics paradigm of evaluating control in simulation [75] is challenging for iBCI given the complexity of simulating these considerations. Understanding how to design evaluation that avoids this open-to-closed loop performance gap remains an open problem for the field, and it is important to note that consistent decoding in FALCON may not necessarily yield consistent real-world control. Nonetheless, FALCON solidifies a current community focus on reducing data requirements. Thus, approaches reaching performance saturation in the FALCON benchmark would significantly advance the field.

FALCON datasets all provide multiple sessions of data for individual subjects. While multi-session data is a substantial advance over single dataset benchmarks (e.g. [46]), methods can have variable performance when applied to different subjects [22, 23]. The relatively unique nature of the tasks in FALCON and the cost of intracortical experiments are currently prohibitive to providing data from the high number of subjects needed to support claims of subject generalization. Evaluation of subject generalization will be an important priority for real-world application when these datasets become more common, and FALCON can be easily adapted to support these datasets.

Finally, FALCON baselines exclusively use spiking activity for decoding. While spiking activity is the default input for many iBCIs, the experimental procedure for determining spiking thresholds often involves researcher discretion. Generally, thresholds are set as a multiple of the RMS of voltages recorded during a baseline period, but the precise multiple and protocol for baseline collection varies from dataset to dataset. To encourage research into stability methods that might avoid human variability or thresholding overall, we have additionally released the raw 30kHz broadband activity for M2 and B1.

**Ethical considerations**    Animal datasets were collected with approval by Institutional Animal Care and Use Committees. Human datasets were collected with Institutional Review Board approval, as part of clinical trials conducted under FDA Investigational Device Exemptions. Informed consent was obtained prior to any experimental procedures. Approvals and experimental procedures can be found in the primary references for each dataset.

FALCON focuses on algorithms that solve a problem specific to iBCIs. Such devices are intended to restore function to individuals with disabilities or impairments resulting from brain injury or disease. However, their widespread adoption raises ethical considerations with respect to the impact of these devices on human identity, privacy, and equity, which are the subject of ongoing study [53, 76].

FALCON also makes use of previously collected animal datasets. Animal models are critical to neuroscientific research that aids in improving our understanding of the brain and develops medical devices for the treatment or assistance of neurological disorders. We hope that by releasing standardized animal datasets, the FALCON benchmarking effort will contribute to the minimization of redundant data collection by allowing researchers to make better use of existing data.

## Acknowledgments and Disclosure of Funding

**Competing Interests**   EMA receives salary from Apple Inc. The MGH Translational Research Center has a clinical research support agreement (CRSA) with Axoft, Neuralink, Neurobionics, Precision Neuro, Synchron, and Reach Neuro, for which LRH provides consultative input. LRH is a co-investigator on an NIH SBIR grant with Paradromics, and is a non-compensated member of the Board of Directors of a nonprofit assistive communication device technology foundation (Speak Your Mind Foundation). Mass General Brigham (MGB) is convening the Implantable Brain-Computer Interface Collaborative Community (iBCI-CC); charitable gift agreements to MGB, including those received to date from Paradromics, Synchron, Precision Neuro, Neuralink, and Blackrock Neurotech, support the iBCI-CC, for which LRH provides effort. CC receives passive royalties from Neuralink and Blackrock Microsystems, and is an inventor of intellectual property optioned by Blue Arbor Technologies. JMH is a consultant for Neuralink and Paradromics, serves on the Medical Advisory Board of Enspire DBS, and is a shareholder in Maplight Therapeutics. JMH is also an inventor of intellectual property licensed by Stanford University to Blackrock Neurotech and Neuralink. VG is the Chief Scientific Officer at Paradromics, Inc. and is a stock options holder at Paradromics, Inc. and Neuralink, Corp. RAG is on the scientific advisory board of Neurowired, has consulted for Blackrock Neurotech, and has received research funding from Blackrock Neurotech. JLC has received funding from Blackrock Microsystems, Inc. CP serves as a consultant to Meta (Reality Labs). These entities did not support this work, did not have a role in the study, and do not have any financial interests related to this work.

**Funding**   This work was supported by: Department of Veterans Affairs, Wu Tsai Neurosciences Institute, NIH-NIDCD U01DC017844, NIH-NIDCD R01DC014034, NIH-NIBIB R01EB028171 (CF); NIH National Institute on Deafness and Other Communication Disorders (grants R01DC018446 and R01DC008358), the NSF Emerging Frontiers in Research and Innovation (EFRI) - Brain-Inspired Dynamics for Engineering Energy-Efficient Circuits and Artificial Intelligence (BRAID) (grant 2223822) and the Kavli Institute for Brain and Mind (IRG no. 2021-1759), "La Caixa" Foundation, and the IIE Fulbright Fellowship (PTM); R21 NS135413-01 (FR); NIH F32HD112173 (SRN); Kavli Institute for the Brain and Mind Innovative Research Grant 2021-1759 and Pew Latin American Fellowship in the Biomedical Sciences (EMA); NSF-NCS Grant 1926576 (CC); Office of Research and Development, Rehabilitation R&D Service, Department of Veterans Affairs (N2864C, A2295R), Wu Tsai Neurosciences Institute, Howard Hughes Medical Institute, Larry and Pamela Garlick, Samuel and Betsy Reeves, Sons Foundation Collaboration on the Global Brain 543045, NIDCD R01-DC014034, NIDCD U01-DC017844, NINDS UH2-NS095548, NINDS U01-NS098968 (JMH); NIDCD R01DC018446 (TQG); NSF EFRI BRAID 2223822 and NIH NIDCD R01DC018446 (VG); NINDS R01 NS079664, NINDS K99/R00 NS101127 (AGR); NINDS UH3NS107714, NINDS U01NS108922, DARPA N66001-10-C-4056 (RAG); the Defense Advanced Research Projects Agency (DARPA) and Space and Naval Warfare Systems Center Pacific (SSC Pacific) under Contracts N66001-10-C-4056 and N66001-16-C-4051 (JLC); NIH-NINDS/OD DP2NS127291 (CP). Any opinions, findings and conclusions or recommendations expressed in this material are those of the authors and do not necessarily reflect the views of DARPA, SSC Pacific, the National Institutes of Health, the Department of Veterans Affairs, or the United States Government.

**Other Acknowledgments**   The authors would like to thank BrainGate clinical trial participant T5, Rehab Neural Engineering Labs clinical study participants, and their families and care partners for their contributions to this research. We would like to thank Beverly Davis, Kathy Tsou, and Sandrin Kosasih for administrative support. We thank Eric Kennedy for animal and experimental support. We thank Gail Rising, Amber Yanovich, Lisa Burlingame, Patrick Lester, Veronica Dunivant, Laura Durham, Taryn Hetrick, Helen Noack, Deanna Renner, Michael Bradley, Goldia Chan, Kelsey Cornelius, Courtney Hunter, Lauren Krueger, Russell Nichols, Brooke Pallas, Catherine Si, Anna Skorupski, Jessica Xu, and Jibing Yang for expert surgical assistance and veterinary care. We thank Marc Schieber for support of original M1 data collection. We thank Gunjan Chhablani and the EvalAI team for support in developing the FALCON evaluation infrastructure.

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

# A Supplementary Material

## A.1 Computational Resources

EvalAI is a platform that allows users to host and participate in AI challenges. It provides a simple interface for the participants to submit their solutions and for the challenge organizers to evaluate them. The link to our FALCON EvalAI page can be found on our challenge website: https://snel-repo.github.io/falcon/

For the FALCON challenge, we provide a container-based image evaluation infrastructure within EvalAI. This environment is primarily based on AWS Elastic Kubernetes Service (Kubernetes Cluster service) but also uses other AWS services like Elastic File System (EFS) to store the ground-truth dataset file(s) on which the evaluation is to be performed. The EFS file system is mounted on the AWS Elastic Compute Cloud (EC2) instance inside the EKS cluster, so the dataset file(s) can be accessed for evaluation.

Once the challenge's participant submits a solution (model) through the EvalAI platform by pushing a Docker container image, an EvalAI agent publishes the solution to the AWS Elastic Container Registry (ECR). The EKS cluster has a single EC2 instance server with pre-defined resources (CPU, memory, storage) which pulls the Docker container image and runs the evaluation script on the submitted solution. The evaluation script is responsible for evaluating the submitted solution and providing a score based on the evaluation criteria. The score is then sent back to the EvalAI platform to report leaderboard metrics.

The EC2 instance type is `g4dn.4xlarge` which has 16 vCPUs, 64 GiB of memory, 1 NVIDIA T4 GPU, and 100 GB of storage. This configuration is sufficient to run the evaluation script on the submitted baseline solutions for the FALCON challenge. EC2 G4dn instances are created to help accelerate machine learning inference and graphics intensive workloads.

## A.2 Interpretation of FALCON Metrics

One goal of the FALCON benchmark is to standardize the metrics used for evaluation of stable decoding performance for each task. As the decoded outputs are very distinct in nature, different metrics were selected for tasks in each domain, each of which is representative of the most widely used metric in each field. As with all benchmarks, metrics should be interpreted with care, as these metrics alone do not necessarily capture all properties of a predicted output. $R^2$ has a convenient maximum at 1, but can be arbitrarily negative if the predictions contain more variance than the ground truth variable. $R^2$ also heavily penalizes predictions that are shifted from the expected center point. WER does not account for how close a given prediction is to the intended word and may overly penalize predictions that are only incorrect by a few characters. MSE can occupy unbounded ranges (i.e., $[0, \infty)$) that can be difficult to contextualize without other relative values. Hence, while models that demonstrate gains in the FALCON metrics will certainly show improved predictions, a poor-scoring model may not necessarily have unreasonable outputs. We recommend visualizing predictions to add additional context to FALCON scores.

## A.3 Datasets

FALCON datasets come from multiple labs and were often collected as parts of larger experiments. Thus, some datasets included in FALCON may share subject and task with other publicly available data, but we ensure that any such releases exclude the specific held-out sessions used in FALCON.

### A.3.1 Data format

All datasets were formatted according to the Neurodata Without Borders (NWB) standard for neurophysiological data. NWB provides open-source Python and Matlab APIs for reading formatted datasets (https://github.com/NeurodataWithoutBorders). The data format builds upon HDF5 and can also be read using any package in any programming language that can access typical HDF5 files. Additionally, FALCON releases a code package that facilitates reading from and analyzing the converted NWB files (https://github.com/snel-repo/falcon-challenge).

### A.3.2 Data hosting and licensing

Datasets are hosted on the Distributed Archives for Neurophysiology Data Integration (DANDI), a platform specifically designed for publishing and sharing neurophysiological data. DANDI generates metadata and identifiers for all uploaded datasets. The datasets we have released on DANDI are distributed under a Creative Commons Attribution 4.0 International license. The authors bear all responsibility in case of violations of rights. The FALCON datasets can be found at the following links:

- M1-A - https://dandiarchive.org/dandiset/000941
- M1-B - https://dandiarchive.org/dandiset/001209
- M2- https://dandiarchive.org/dandiset/000953
- H1- https://dandiarchive.org/dandiset/000954
- H2- https://dandiarchive.org/dandiset/000950
- B1- https://dandiarchive.org/dandiset/001046

### A.3.3 Dataset documentation - `M1`

**General description**    This dataset contains unsorted spike times and electromyography (EMG) data from two macaques performing a reach and grasp task. Neural activity was recorded from Floating Microelectrode Arrays (FMAs; Microprobes) implanted in the motor cortex (M1). EMG was recorded from 16 muscles of the right hand and upper extremity. EMG electrodes were comprised of 32-gauge, Teflon-coated, multi-stranded, stainless steel wire. They were implanted in bipolar pairs, separated by 5-10mm along the axis of the muscle. Muscle targeting and separation were performed as described in [77]. Wires were tunneled subcutaneously to exit the skin of the back at the midline in four separate bundles. Each bundle ended in a separate connector sewn into the back of a jacket worn by the monkey.

Our release of `M1-A` consists of 4 held-in datasets spanning 5 days, each with 53-61 minutes of calibration data, and 3 held-out datasets spanning 21 days, which have only 1.1-2.2 minutes of calibration data available. We also release a second monkey, `M1-B`, for which there are 4 held-in datasets spanning 7 days (with 52-60 minutes of calibration data each) and 4 held-out datasets spanning 7 days (with 0.9-1.3 minutes of calibration data).

**Source**    This dataset was collected by Adam G. Rouse with the support of Marc Schieber. The data was collected for the purpose of to dissociating the effects of location and object during reach-to-grasp behaviors. The experiment and data collection is described and features in a number of papers, including [54–57]. The dataset creators have granted permission to use and distribute the dataset sessions as part of the benchmark.

**Intended use**    This dataset has been curated for evaluating stable decoding approaches as part of the FALCON benchmark. Muscle activations are a target for iBCIs through functional electrical stimulation [59] and are important scientifically for motor control as they are the direct output of commands sent from the central nervous system. The dataset is available on DANDI to allow others to evaluate their approaches on the data.

**Experimental design**    The experimental task is to reach to, grasp, and manipulate 4 different objects at 8 different locations, arranged in a center-out fashion. The 4 objects are separated by 45 degrees with a fifth object in the middle. The center object is a coaxial cylinder, and the four peripheral (target) objects include: a button mounted inside a tube, a sphere, a perpendicular cylinder, and a coaxial cylinder identical to the center object. The trained manipulation schemes are as follows: cylinders are pulled towards the subject, the button is pushed, and the sphere is rotated 45 degrees.

The objects are arranged in a fixed order (perpendicular cylinder, coaxial cylinder, button, sphere) spanning 135 degrees of a circle. Objects were rotated to one of eight orientations in 22.5 degree increments (some positions excluded due to biomechanical or visual constraints). This leads to a total of 8 possible locations per object. Trials begin with the monkey pulling on the center cylinder and holding for 1500-2000ms. A blue light cues a pheripheral object, which the monkey needs to reach

to, grasp, and manipulate. For a trial to be successful, the monkey complete these interactions with the cued object within 1000ms and hold the object in the manipulated state for 1000ms.

**Data collection methods**   The dataset contains neural activity of two rhesus monkeys implanted with 6 FMAs, 16 channels each, electrodes of length 1.5-8mm. We focus on 4 of these arrays (H, I, J, and K) placed in primary motor cortex (m1) that were consistently recorded from throughout the duration of this dataset. Recordings were sampled at 30kHz. Thresholds to extract spiking data were manually set for each channel and may vary from session to session. Intramuscular EMG was recorded from 16 muscles including: anterior deltoid (DLTa), posterior deltoid (DLTp), pectoralis major (PECmaj), short head of biceps (BCPs), lateral head of triceps (TCPlat), flexor carpi radialis (FCR), flexor carpi ulnaris (FCU), extensor carpi radialis brevis (ECRB), extensor carpi ulnaris (ECU), radial and ulnar flexor digitorum profundus (FDPr, FDPu), abductor pollicis longus (APL), extensor digitorum communis (EDC), thenar muscle group (Thenar), first dorsal interosseus (FDI), and hypothenar muscle group (Hypoth).

**Processing**   For all sessions, thresholds crossings were computed in 20ms bins. We apply a standard set of preprocessing for the EMG data decoding target as follows: notch filter the signal at 60Hz and harmonics to remove line noise, high pass filter (acausal Butterworth filter, 4th order) with a 65Hz cutoff, rectify the resulting signal, clip the signal at the 99th quantile, scale the signal at the 95th quantile, resample to 50Hz, rectify again, and finally low pass filter (acausal Butterworth filter, 4th order) with a cutoff of 10Hz. In both held-in and held-out files, the last 40% of the data was reserved for evaluation. Remaining data is released for held-in sessions for calibration. Ten trials were released for each held-out calibration set.

### A.3.4   Dataset documentation - `M2`

**General description**   This dataset contains unsorted spike times and finger kinematics from a macaque performing an individuated finger control task. Neural activity was recorded from precentral gyrus. Finger position and finger velocity were also recorded.

`M2` includes 4 held-in datasets over 10 days, with between 5.9-13.3 minutes of calibration data per session available, and 4 held-out datasets over 26 days with between 0.8-1.7 minutes of calibration data provided.

**Source**   This dataset was collected by Samuel R. Nason-Tomaszewski, Matthew J. Mender, and Cynthia A. Chestek at the University of Michigan. The data was collected to study closed-loop individuated finger control with an iBCI. The experiment and data collection are discussed in [60]. The dataset creators have granted permission to use and distribute the dataset sessions as part of this benchmark.

**Intended use**   This dataset has been curated for evaluating stable decoding approaches as part of the FALCON benchmark. Dexterous hand control is an important behavior that iBCIs aim to restore, and robust decoding approaches that work well with individuated finger movement behaviors may aid in this endeavor. The dataset is available on DANDI to allow others to evaluate their approaches on the data.

**Experimental design**   The monkey is shown a virtual hand whose finger state mirrors that of a manipulandum. The manipulandum was designed to measure the monkey's finger state. Only two independent degrees of freedom are allowed in the manipulandum - the index finger and MRS group, which bundles middle, ring, and small fingers.

In trials, the monkey is shown visual cues (colored dots) to indicate a target finger state, and the monkey moves their fingers to the cued positions. The visual cues are colored corresponding to the colors of the fingers, indicating which finger group should be moved to each target. The monkey is trained to proficiency, and each trial is on the order of a second.

The finger and target positions are bounded to the range [0, 1], where 0 represents full extension of the finger group, and 1 represents full flexion of the finger group. The task begins with both targets at a central position (0.5), and the targets return to the central position every other trial (i.e. a typical center-out-and-back task paradigm). In between central targets, the task randomly selects from a set

of target positions that moves one or both groups of fingers away from the central position. Whether one or both groups of fingers moves in a given trial is randomly selected, and the distance to which each group moves is also randomly selected (±0.2, ±0.3, or ±0.4 from the central position). For trials in which both finger groups are instructed to move from the central position, the magnitude of instructed movement for both finger groups is always equal, but the direction of movement is not always the same. There are no trials in which one group was instructed to move to +0.4 and the other group was instructed to move to -0.4 due to the difficulty in separating fingers that far.

**Data collection methods**   The M2 dataset contains neural activity of a rhesus macaque with two 64-channel Utah microelectrode arrays placed in the precentral gyrus, of which 96 channels are are provided. Recordings were originally sampled at 30kHz. Finger positions and velocities were recorded at 1kHz using actuator velocities measured by the manipulandum.

**Processing**   For all sessions, we release threshold crossings in 20ms bins. We similarly resample the finger kinematics to 50Hz for consistency. In both held-in and held-out sessions, the last 40% of the trials were reserved for evaluation splits. The preceding 60% of held-in session data is released for calibration. For held-out sessions, the first 10% of each session is released as the calibration split.

### A.3.5   Dataset documentation - H1

**General description**   This dataset contains unsorted spike times and robotic actuator kinematics from a human iBCI participant during an open-loop attempted reach-to-grasp task. Neural activity was recorded from motor cortex. Behavioral covariates include the 7-degree-of-freedom kinematics corresponding to each cue. The 7 degrees of freedom are: 3 dimensions of translation, 1 dimension of rotation (roll), and 3 dimensions for grasp shaping. The grasp shaping dimensions are: pinching (flexion and extension of thumb/index/ring finger), scooping (flexion and extension of ring and pinky finger), and thumb abduction.

H1 consists of 6 held-in sessions over 20 days (5-9 minutes of calibration data each) and 7 held-out sessions over 15 days (1.5-1.8 minutes of calibration data provided).

**Source**   This data was collected by Sharlene Flesher, John Downey, Jennifer L. Collinger, and Robert A. Gaunt at the University of Pittsburgh as part of a clinical trial of iBCIs for sensorimotor control. The experiment and data collection protocol are described in [1, 63, 64]. Participant and date-time information have been obfuscated for deidentification. This data was collected under an Investigational Device Exemption from the U.S. Food and Drug Administration and is registered at ClinicalTrials.gov (NCT01894802). The study was also approved by the Institutional Review Boards at the University of Pittsburgh and the Space and Naval Warfare Systems Center Pacific. Informed consent was obtained before any study procedures were conducted and included permission for data sharing. Dataset collectors granted their permission to use and distribute these sessions as part of this benchmark.

**Intended use**   This data has been curated for evaluating stable decoding approaches as part of the FALCON benchmark. The H1 dataset provides an example of a high degree-of-freedom behavior, which may pose specific challenges to robust decoding approaches. The dataset is available on DANDI to allow others to evaluate their approaches on the data.

**Experimental design**   As this is an open loop task, the participant is asked to attempt to perform a movement cued with a virtual arm. The virtual arm movement occurs in phases: reach, grasp, carry, release. Each phase begins with a presentation of a combined visual and word cue for a particular movement, so the participant can prepare an imagined movement, and an another cue to execute the imagined movement. The participant has had practice following these cues to calibrate similar decoders before this dataset was collected.

**Data collection methods**   Neural data was collected at 30kHz from two Utah arrays placed in the hand and arm region of motor cortex.

**Processing**   For all sessions, we preprocess the neural data into 20ms bins. Robotic arm and hand kinematics were also downsampled to 50Hz to be consistent with the neural data. For both held-in

and held-out sessions, we reserve the last 20% of each file for evaluation. All remaining data is released for held-in sessions, to be used for calibration. The first 20% of each file is released for each held-out session as the calibration split.

### A.3.6 Dataset documentation - H2

**General description**   H2 contains unsorted spike times and sentence prompts from a of a BrainGate2 pilot clinical trial participant (referred as T5) during an open-loop attempted handwriting task. Neural activity was recorded from the hand "knob" area of arrays placed in the precentral gyrus. Behavioral covariates for this task are not continuous but rather the cued sentence for each trial.

H2 consists of 21 held-in sessions over 287 days (7-45 minutes of calibration data each) and 5 held-out sessions over 176 days (1-2 minutes of calibration data provided).

**Source**   This data was collected by Chaofei Fan, Leigh R. Hochberg, and Jaimie M. Henderson at Stanford University as part of the BrainGate2 Neural Interface System clinical trial (ClinicalTrials.gov Identifier: NCT00912041, registered June 3, 2009) on iBCIs. The experiment and data collection protocol are described in [5, 29]. This pilot clinical trial was approved under an Investigational Device Exemption (IDE) by the US Food and Drug Administration (Investigational Device Exemption G090003). Permission was also granted by the Institutional Review Boards of Stanford University (protocol 20804). Informed consent was obtained prior to any study procedures being conducted. Dataset collectors granted their permission to use and distribute these sessions as part of this benchmark.

**Intended use**   This dataset has been curated for evaluating stable decoding approaches as part of the FALCON benchmark. Restoring communication is a primary high-level goal of iBCIs, and the handwriting task is a prime example of a brain-to-text decoding scheme that is appropriate for participants with arrays placed in motor areas. The dataset is available on DANDI to allow others to evaluate their approaches on this data.

**Experimental design**   On each trial, T5 was prompted to copy a cued sentence by writing individual characters (26 English letters, 4 punctuations, and a special ">" character to represent space). T5 was instructed to attempt to write as if his hand were not paralysed, while imagining that he was holding a pen on a piece of ruled paper. Each trial has a fixed-length delay period and a variable-length go period. During the delay period, T5 reads the cued sentence. Once the go cue is on, T5 starts to copy the sentence letter by letter and uses a verbal cue to indicate he has finished copying. In each session, both open-loop and "pseudo-closed-loop" trials were collected. In open-loop trials, the participant only sees the cued sentence; in pseudo-closed-loop trials, the real-time decoded letters are shown.

**Data collection methods**   Data was collected from two 96-channel intracortical Utah arrays placed in the hand "knob" area of the participant's left hemisphere precentral gyrus. Original neural recordings were collected at 30kHz.

**Processing**   For all sessions, neural data was binned at 20ms. We provide only data from within each trial period, as inter-trial time could be long and lead to unnecessarily large data files. Each trial is accompanied by the cued sentence. For both held-in and held-out sessions, 40% of the trials are reserved for evaluation. For held-in sessions, remaining trials are released for calibration. For held-out sessions, only 3 trials are released for few-shot calibration.

### A.3.7 Dataset documentation - B1

**General description**   The B1 dataset contains unsorted spike times, audio recordings, and audio spectrograms from a zebra finch songbird during natural vocal behavior. Neural activity was recorded using Neuropixels 1.0 probes from the motor region robust nucleus of the arcopallium (RA). The dataset includes a precomputed spectrogram derived from the recorded amplitude waveform corresponding to each vocal epoch, which constitutes the decoding target for this dataset. Due to the inherent non-determinism in spectrogram computation, we include the computational function used to derive the spectrograms. Additionally, the raw amplitude waveform is available for use in decoding strategies that might prefer it as a target before conversion to spectrogram form.

B1 consists of 3 held-in sessions corresponding to 3 consecutive days of recordings, with 7-26 seconds of calibration data per session available, and 3 held-out sessions recorded over the following 6 days, with 2.7 seconds of calibration data per session provided.

**Source**   This dataset was collected by Pablo Tostado-Marcos, Ezequiel Matias Arneodo, Timothy Q. Gentner, and Vikash Gilja. The original experiment and procedures are described in [67]. Data collectors granted their permission to use and distribute these data as part of this benchmark.

**Intended use**   This dataset has been curated for evaluating stable decoding approaches as part of the FALCON benchmark. The dataset offers an alternative to brain-to-text iBCIs by focusing on the direct reconstruction of audio spectrograms from neural signals, thereby encouraging decoding approaches that preserve the prosodic elements of vocal behavior. The B1 dataset proposes the songbird model as a proxy for human vocalization, supporting the development of neuroprostheses aimed at restoring communication capabilities and advancing stable-decoding methods. This dataset is available on DANDI to allow others to evaluate their approaches on these data.

**Experimental design**   An adult, male zebra finch songbird was implanted with a single high-density Neuropixels 1.0 probe targeting the telencephalic motor region RA in the right brain hemisphere. Simultaneous neural and behavioral (song) data were collected in a single-housing acoustically-isolated chamber during awake-singing. The bird was allowed to move and sing freely during 120-240 minute-long recording sessions.

**Data collection methods**   Neural data and vocal behavior were collected simultaneously. Voltage signals were recorded by 384 Neuropixels channels, amplified, band-pass filtered (300Hz-10000Hz), multiplexed and digitized at 30kHz on the Neuropixels headstage, and transferred to the data acquisition module. We focus on 85 channels in the B1 dataset corresponding to region RA. Audio signals were recorded at 25kHz and high-pass filtered (250Hz) for subsequent extraction of spectral features. The stereotypy characteristic of zebra finch song enabled the segmentation of non-overlapping sequences of song syllables, or motifs, composing the bird's own song. These motifs are similar in their syntactic structure but vary in timing, pitch and syllable count across vocal renditions. Custom software was used for extracting song motifs from the audio recordings and for computing the spectrogram representations to be used as decoding targets.

**Processing**   To allow maximum flexibility in decoder design, we provide threshold crossing activity at the original 30kHz resolution. Thresholds were independently set on a per-channel basis and may vary across sessions. The amplitude waveform corresponding to each motif rendition, synchronized to neural data, is also provided at the original 25kHz sampling rate. The audio signals provided were band-pass filtered within the relevant birdsong frequency range (250Hz-8000Hz) and de-noised using the *noisereduce* Python package [78]. The spectrogram corresponding to each amplitude waveform, which constitutes the ultimate decoding target, is provided at 1kHz resolution. We provide only a window of data around each 700ms-long motif (100ms before motif onset and 100ms after the end of the motif; total epoch length is 900ms). For both held-in and held-out datasets, 40% of the song motifs are reserved for evaluation. On held-in datasets, the remaining 60% of the data is released for calibration. On held-out datasets, 3 motifs are made available for the few-shot calibration split.

## A.4   Baseline Implementation Hyperparameters

### A.4.1   Wiener Filter (WF)

**Description**   For the movement datasets, linear static and oracle decoders are done using a Wiener filter. Wiener filters predict the current value of an output signal using previous timesteps, as defined by:

$$y[t] = \sum_{i=0}^{I-1} w_i x[t-i]$$

where $y[t]$ is the output signal at time $t$, $x[t]$ is the input signal at time $t$, $w_i$ is the filter coefficient, and $I$ is the number of previous samples to use for decoding. In our decoder, the input signal $x$ is

the smoothed neural data, $y$ is the behavioral output to predict, and $I$ is the number of time bins of history. The weights are fit using a matrix formation of the above equation:

$$W = (X^T X + \lambda I)^{-1} X^T y$$

where $W$ is a matrix of filter coefficients, $X$ represents the predictor data with history and bias, and $y$ represents the output signal. $\lambda I$ represents a diagonal matrix with the L2 regularization constant filling the diagonal. The bias term is not regularized and therefore its diagonal entry is set to zero. The L2 regularization aims to avoid decoder overfitting by penalizing solutions with large individual weights.

**Implementation**  We implement the WF as a Ridge Regression model using the Scikit-learn library [79].

**Parameter Optimization**  L2 regularization values are obtained using 5-fold cross validation. We sweep a range of 20 values spanning 1e-5 to 1e5 in logspace. For each value, we train and test a Wiener filter using 5-fold cross validation, testing the decoder on a held-out fold. The optimal regularization value was selected based on which value yielded the highest performance metric. Final performance was reported on the held out fold.

To determine the number of bins of history to use for each dataset, we swept from $I = 0$ to $I = 30$ by training on all held-in calibration and held-out oracle data splits and reporting performance on the evaluation split for each dataset, separately. We selected the appropriate number of bins by plotting the resulting $R^2$ for each session and choosing the value that coincided with the elbow for the most sessions within a dataset. These results are shown in **Figure 5**. We ultimately selected $I$=30 bins (600ms) for M1, $I$=7 bins (140ms) for M2, and $I$=30 bins (600ms) for H1.

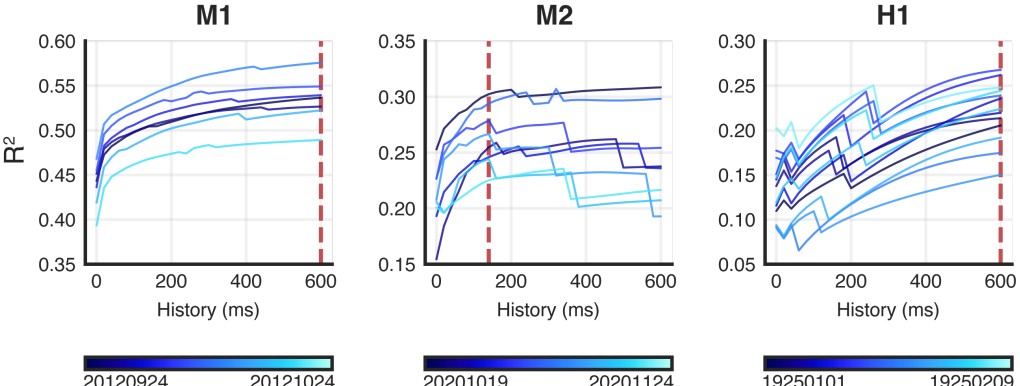

**Figure 5. Sweeps to determine WF history**. We train WF decoders on each session (differentiated by shades of blue, see colorbar) of all datasets with history from $I = 0$ms to $I = 600$ms and evaluate the prediction $R^2$ on the held-out split. We select the history that maximizes performance for most sessions, indicated with the vertical red dashed line. For M1 and H1, performance continued to rise with increased history. We capped history at 600ms to ensure reasonable use for iBCIs [80, 81], but selected this maximal value for these datasets. For M2, performance reached an elbow with 140 ms of history.

**Code Availability**  The WF decoder used for FALCON is available on our Github repository at: https://github.com/snel-repo/falcon-challenge/blob/main/decoder_demos/sklearn_decoder.py

### A.4.2 RNN Decoder - Movement

**Description**  The RNN baseline uses a 1-layer LSTM followed by a linear readout of behavior. It is a simple supervisd baseline.

**Implementation**  This minimal baseline excludes any multi-session layers, and thus was only trained on single sessions of data to report oracle and zero-shot results. It is implemented within the NDT2 codebase, but uses simple Pytorch layers.

**Parameter Optimization**  We sweep learning rate and model hidden size, though find training varies little with these choices. We sweep 6 parameter combinations per model. Models train very quickly (2-10 minutes on a single 2080 NVIDIA GPU).

**Code Availability**  RNN baseline code is available in the NDT2 codebase: https://github.com/joel99/context_general_bci/.

### A.4.3  Neural Data Transformer 2 [32]

**Further results**  We present additional NDT2 baselines in **Table 2**.

**Table 2. FALCON movement baselines with zero-shot and static NDT2**. We include NDT2 zero-shot and static results for comparison with the RNN baseline. Oracle single-session models and zero-shot transfer achieve comparable results with the vanilla RNN decoder.

| | Movement (Held-Out $R^2$ / Held-In $R^2$ $\uparrow$) | | | |
| --- | --- | --- | --- | --- |
| | Class | M1 | M2 | H1 |
| Wiener Filter (WF) | OR | $0.53_{\pm 0.04}/0.54$ | $0.26_{\pm 0.03}/0.27$ | $0.21_{\pm 0.04}/0.24$ |
| RNN | OR | $0.75_{\pm 0.05}/0.75$ | $0.56_{\pm 0.04}/0.59$ | $0.44_{\pm 0.13}/0.51$ |
| NDT2 | OR | $0.71_{\pm 0.06}/0.72$ | $0.44_{\pm 0.08}/0.53$ | $0.38_{\pm 0.13}/0.44$ |
| NDT2 Multi | OR | $0.78_{\pm 0.04}/0.77$ | $0.58_{\pm 0.04}/0.62$ | $0.63_{\pm 0.08}/0.68$ |
| WF | ZS | $0.34_{\pm 0.06}/0.46$ | $0.06_{\pm 0.04}/0.15$ | $0.16_{\pm 0.03}/0.20$ |
| RNN | ZS | $-.61_{\pm 0.48}/0.51$ | $0.13_{\pm 0.09}/0.17$ | $0.08_{\pm 0.17}/0.29$ |
| NDT2 [32] | ZS | $0.11_{\pm 0.11}/0.55$ | $-0.03_{\pm 0.15}/0.28$ | $0.10_{\pm 0.10}/0.32$ |
| NDT2 Multi [32] | FSS | $0.59_{\pm 0.07}/0.77$ | $0.43_{\pm 0.08}/0.63$ | $0.52_{\pm 0.04}/0.62$ |

**Description**  NDT2 is a Transformer-based deep neural network previously demonstrated to enable multi-context neural data modeling with or without any specific parameters for different datasets. The model tokenizes each timestep of the input data into contiguous subsets of fixed length along the full channel dimension. As there are multiple tokens per timestep, the multiple input tokens must be merged to produce per-timestep decoding. In this work, the NDT2 baselines use cross-attention for behavior decoding. Additionally, the models perform neural data reconstruction as a secondary objective.

**Implementation**  NDT2 models are prepared as the other single-session baselines, i.e. separate models are trained per session and the model that performs best on held-in data is used for zero-shot transfer. NDT2 Multi models use all available calibration data at once, e.g. for the oracle NDT2 Multi model, a single model was trained with all held-in calibration, held-out calibration, and held-out redacted data; the regular NDT2 Multi model uses all calibration data. Thus NDT2 does use the few-shots of calibration data available in future sessions for predicting an early held-out session; this is an implausible design for real-world use, chosen for simplicity.

As described in Section A.5.1, the training dataloader was modified to provide random fixed length subsets over either training on trialized data or direct splitting of the continuous data. This is required on M1 and M2 for the model to be more robust to FALCON's continuous evaluation.

**Parameter Optimization**  NDT2 used a fixed grid search on model parameters and learning rate for each of M1, M2, H1, and trained with early stopping. The checkpoint with best validation score was used to compute the baseline metric. Individual runs (6 runs per sweep) cost up to 3 hours per model (for M1 data) on a single NVIDIA 2080 GPU.

**Code Availability**  The codebase and checkpoints, along with the specific hyperparameter sweeps, are on the public Github repo: https://github.com/joel99/context_general_bci/.

### A.4.4  NoMAD [22]

**Description**  Nonlinear Manifold Alignment with Dynamics (NoMAD) is a manifold alignment decoder stabilization approach that operates in a few-shot unsupervised regime. NoMAD frames the stabilization problem using pairs of datasets consisting of an initial calibration dataset, "Day 0," where

both neural and behavioral data are available, and a later dataset that contains neural nonstationarities with respect to Day 0, termed "Day K," for which only neural data is available.

On Day 0, a dynamics model is trained, and a decoder is trained to map from the inferred dynamics to the behavior. The model and decoder are then frozen. On Day K, using neural calibration data, a feedforward alignment network is trained to map the new neural data onto the fixed dynamics model. This alignment network is trained primarily using a Kullback-Leibler divergence cost between the RNN states of the dynamics model on Day 0 and Day K. After alignment network training, the fixed decoder can be applied to the Day K dynamics to maintain accuracy.

**Implementation**   The Day 0 dynamics model is latent factor analysis via dynamical systems (LFADS) [82, 17]. Unlike the model with the behavioral readout described in [22], the LFADS models here are the standard nonautonomous models described in [17, 83, 84]. Of note, we select only one Day 0 dataset from the held-in sessions for each task; for M1 , this is the `20120926` session, for M2, `2020-10-19-Run2`, and for H1, `19250113T120811`. The remaining details are consistent with [22]. The decoder applied is a Wiener Filter with 4 bins of history for M1and M2; for H1, we used 4 bins of history for most held-in sessions and 8 bins of history to evaluate the Day 0 session that NoMAD aligned to as well as all held-out sessions.

**Parameter Optimization**   Day 0 LFADS model parameters were optimized using AutoLFADS [83, 84]. AutoLFADS was trained using 8 NVIDIA GeForce RTX 2080 Ti GPUs with training time less than one hour per held-in dataset. NoMAD alignment network hyperparameters were optimized using a random search, with each model in the random search using one NVIDIA GeForce RTX 2080 Ti GPU (training time approx. 20 minutes). Wiener Filter bins of history were optimized using a grid search. Models and decoders were selected based on those which had the highest accuracy on the held-out calibration data.

**Code Availability**   Because [22] is still under review, the Systems Neural Engineering Lab will not release the code right now. The code will be made available when this paper has been published.

### A.4.5   CycleGAN  [23]

**Description**   Cycle-Consistent Adversarial Network (CycleGAN) is a decoder stabilization approach based on the use of Generative Adversarial networks (GANs) that, like NoMAD, operates in a few-shot unsupervised regime. Similar to a conventional GAN, CycleGAN architecture consists of a pair of neural networks, a generator and a discriminator. The generator (or aligner) is trained to transform the high-dimensional neuronal firing rates from the calibration "Day K" dataset into a form resembling the initial "Day 0" dataset, which was used to train the fixed neural-to-behavior decoder. The discriminator is trained adversarially to the generator to maximize the distance between the distributions of Day K and Day 0 datasets. Unlike regular GANs, CycleGAN also implements a cycle-consistent loss that regularizes the learning of the Day K to Day 0 mapping function, thereby reducing the search space and making training more stable. This is achieved by adding a second pair of generator and discriminator networks that are trained to learn the opposite transformation (i.e., from Day 0 to Day K). After CycleGAN training, the aligner is used to transform Day K into Day 0 data to maintain the accuracy of a fixed Day 0 decoder.

**Implementation**   Here we select only one Day 0 dataset from the held-in sessions for each task to compute the fixed decoder and the subsequent Day 0 to Day K CycleGAN aligners. (`20120926` session for M1 , `2020-10-28-Run1` for M2, and `19250120` for H1). The decoder used is a Wiener Filter with 8 bins of history for all the tasks.

**Parameter Optimization**   Training CycleGAN is computationally efficient since the generator and discriminators pairs are feedforward neural networks. The training process can be completed on any modern CPU in under two minutes. We used the same hyperparameters as described in [23]. The selection of CycleGAN aligners and decoders was based on those that achieved the highest accuracy on the held-out calibration data.

**Code Availability**   A step-by-step tutorial on the use of Cycle-GAN for neural alignment can be found on the public Github repo: `https://github.com/limblab/adversarial_BCI/blob/main/Cycle_GAN_aligner.ipynb`.

### A.4.6  RNN Decoder and Language Models - Communication (H2)

**Description**   The baseline RNN decoder for H2consists of a shared Gated Recurrent Unit (GRU) backbone and a set of session-specific affine transform layers. The neural activity first undergoes an affine transformation via the session-specific layer, and then the GRU backbone decodes the transformed neural activity into characters. The GRU and session-specific transforms are trained end-to-end on multiple sessions.

The language model (LM) evaluates a sentence's probability. Given the character probabilities output from RNN, we use beam search and an LM to find the most likely sentence.

**Implementation**   The baseline RNN is a 2-layer GRU with 512 hidden units. A softmax layer maps the GRU outputs to character probabilities. The session-specific layer is implemented as a linear layer with the same number of units as the input neural features.

The baseline RNN is trained jointly on multiple sessions to maximize performance. The zero-shot/static model is trained on calibration splits of all held-in sessions. The oracle model is trained on oracle splits of held-out sessions (not including sessions later than the testing day) plus the calibration splits of all held-in sessions.

We use a 3-gram LM trained on the OpenWebText2 corpus to convert the RNN outputs into words in real-time. To further improve the accuracy, we use GPT2-XL to rescore the outputs from the 3-gram LM. See details in [29].

**Parameter Optimization**   Hyperparameters were optimized by grid search in [29]. No additional optimization is done for this work.

**Code Availability**   Code and pre-trained models are available here: `https://github.com/cffan/` `CORP`

### A.4.7  CORP [29]

**Description**   CORP leverages language models (LMs) for test-time adaptation. It first uses LMs to correct errors due to nonstationarity in decoded sentences. The corrected sentences are used as pseudo-labels to calibrate the RNN. The calibration runs after the user finishes writing a sentence.

**Implementation**   We use the same RNN and LM implementations as in A.4.6. For FALCON, we first trained a seed model on calibration splits of held-in sessions. Then, for each held-out evaluation session, we take the calibrated model from the previous session (seed model for the first evaluation session), use the model to decode trials from the new sessions, and run calibration after decoding.

**Parameter Optimization**   Hyperparameters were optimized by grid search in [29]. No additional optimization is done for this work.

**Code Availability**   Code and pre-trained models are available here: `https://github.com/cffan/` `CORP`

### A.4.8  EnSongdec [67] - B1

**Description**   The EnSongdec model is a brain-to-song deep neural network that enables synthesis of an amplitude waveform from input brain activity. The model features a feed-forward neural network (FFNN) trained to predict audio (song) embeddings, $z$, from each timestep of input neural data. The FFNN is coupled to a Quantizer-Decoder network extracted from a pre-trained, state-of-the-art audio codec (EnCodec) [68]. The quantization layer converts $z$ into a compressed latent representation, $z_q$, utilizing residual vector quantization (RVQ). The decoder network uses transposed convolutions to reconstruct the time-domain audio signal at its original sampling rate.

**Implementation**   We first used the Encoder network of a pre-trained EnCodec model to extract meaningful embedding representations of birdsong. To prioritize reconstruction quality over data compression and streamability, we opted for minimal EnCodec audio compression settings (24kbps at a 48kHz upsampled input). Next, we optimized custom feed-forward neural networks to translate

input neural signals into continuous embedding representations of birdsong. Threshold-crossing inputs were smoothed using a 1-d Gaussian kernel ($\sigma = 30$) and downsampled to match the rate of the target song embeddings (150 samples per second; 7ms bins). 14ms of neural data (2 bins) were used to predict each audio embedding sample. The resulting feed-forward neural network featured an input layer of size $i = N \times history\_bins$ (where $N = 85$ denotes the number of Neuropixels channels targeting the brain region of interest), two 64-unit hidden layers and a 128-unit output layer corresponding to the dimensionality of the embedding space. ELU activation functions were employed and mean square error (MSE) was used as the reconstruction loss to train FFNNs. The Quantizer-Decoder network excised from the aforementioned pre-trained EnCodec model was coupled to the FFNN to synthesize a continuous time-domain song signal. The spectrogram of the reconstructed song was compared to the spectrogram of the original recorded song to evaluate the performance of EnSongdec.

**Parameter Optimization**    Hyperparameters were optimized using a grid search approach based on minimal song reconstruction error. We used Weights & Biases for experiment tracking. Models were trained on a proprietary cluster of servers using NVIDIA GeForce RTX 2080 Ti GPUs.

**Code Availability**    Code to implement EnSongdec can be found in the public Github repository: https://github.com/pabloslash/EnSongdec

## A.5    Evaluation Parameters

### A.5.1    Continuous vs Trialized Evaluation for Motor Tasks

Real-world BCIs will require continuous decoding of user intention. This motivated us to design FALCON evaluation to be continuous, despite the fact that the evaluation data was often collected in a trialized setting. When making this design choice, we identified a sensitivity to training and evaluation context length that varied across datasets. Specifically, we trained trialized NDT2 decoders, where training data is divided into single behavioral trials, and continuous NDT2 decoders, where the continuous training data is divided into fixed length segments. Trialized models used up to 4 seconds of history (M1 and H1 had trials in excess of this length, but M2 trials averaged to about 1 second. Further, continuous models did not directly train on data split into segments of a given length, but rather required augmentation. Data was split into segments longer than the target length; e.g. for 2 seconds of history, data might be split into 4 second segments, and a 2 second slice was drawn at random. Decoders of either kind were evaluated in a trialized setting, where signals about trial change could be used to reset model input, or in a continuous setting, where these signals were not available.

**Fig. 6** illustrates the sensitivity of trialized models to continuous evaluation. Trialized models often performed well in trialized evaluation, and in particular did not degrade with long histories. However, when evaluated in continuous settings, performance dropped precipitously in M1 and M2. In H1, models trained on trialized data continue to improve with higher histories, on either trialized or continuous evaluation.

In contrast, models trained on continuous data fail with long histories, but peak performance was often comparable to peak trialized performance. Accordingly, NDT2 baselines for M1 and M2 trained with continuous data and with trialized data for H1.

The dependence of trialized training on trialized evaluation suggests that models are exploiting trial structure (distinct behavior at the start, middle, and end of trials) to reduce uncertainty about decoding at different timepoints. This is likely to not benefit closed loop control, where decoding should be able to produce flexible behavior at any timepoint. Since continuous models are also able to achieve similar performances, it is possible that continuous models are still exploiting trial structure by inferring the part of the trial that needs to be currently decoded. Moreover, the fact that performance continues to improve with longer context for H1 remains a particularly concerning edge case that may be exploited in FALCON leaderboards. We do not restrict H1 context in evaluation for simplicity, but encourage works to report the history they use as input for context. We also encourage work that sheds light on why H1 behaves differently than M1 and M2.

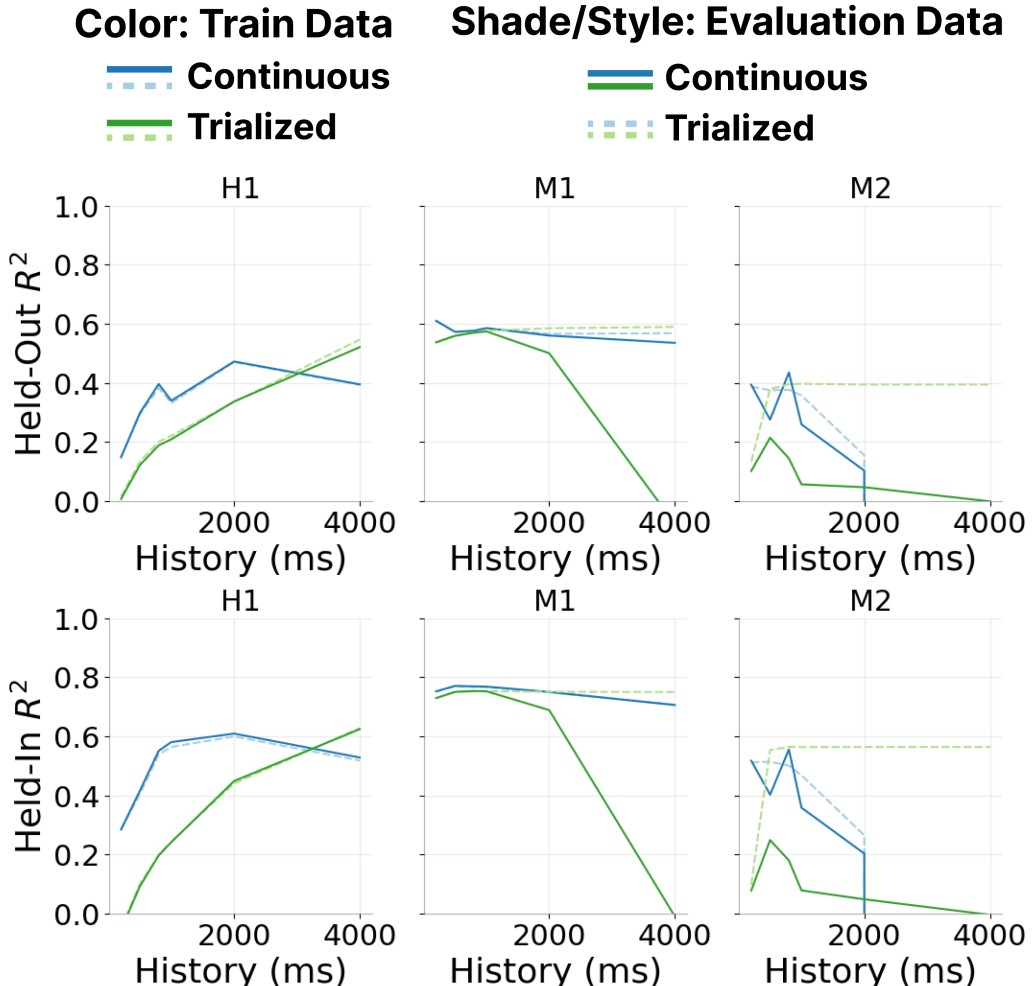

**Figure 6. Continuous vs Trialized Decoding**. We train and evaluate NDT2 decoders on combinations of trialized and continuous data. History indicates the length of context the model uses to make predictions during evaluation. These decoders are sensitive to data trialization, with varying effects across datasets. Models trained with trialized data in M1 and M2 fail significantly when evaluated in a continuous fashion (indicating dependence on trial structure during decoding). Continuous models can often match the performance of trialized models in either trialized or continuous evaluation, but is sensitive to the length of history used. H1, unlike the other datasets, sees continued gains with increased context beyond the explored range even under continuous evaluation, indicating further possibility of model exploitation of trial structure. Held-in metrics show similar trends. FALCON is susceptible to models that exploit these gains.

### A.5.2 Determining data volumes for few-shot calibration on held-out days

FALCON aims to enforce the realistic constraint that calibration on new sessions will have limited neural and behavioral data. To ensure that the few-shot problem was well-represented, we established that the held-out calibration splits were insufficient to train new linear decoders on their own. As shown in **Table 3**, performance of WF decoders on the movement datasets trained on the held-out calibration splits under-performs those trained on the held-out oracle splits.

**Table 3. WF decoder performance on held-out data splits.** WF decoders trained on held-out calibration and held-out oracle splits for all movement datasets.

| Training Data | M1 | M2 | H1 |
|---|---|---|---|
| Held-out calibration | $0.24_{\pm 0.04}$ | $0.14_{\pm 0.05}$ | $0.11_{\pm 0.03}$ |
| Held-out oracle | $0.53_{\pm 0.04}$ | $0.26_{\pm 0.03}$ | $0.21_{\pm 0.04}$ |

