# OpenReview forum: "Few-shot Algorithms for Consistent Neural Decoding (FALCON) Benchmark"
_NeurIPS.cc/2024/Datasets_and_Benchmarks_Track — NeurIPS 2024 Track Datasets and Benchmarks Poster_

### Official Review · Reviewer_pDQb · 2024-07-19
**Interesting work in a under-explored direction, but improvements needed**

**Rating:** 6
**Confidence:** 4

**Review:**

For lists of pros and cons, please see "Strengths" and "Opportunities For Improvement" sections respectively.

**Strengths:**

This paper makes a decent effort in an underexplored direction, i.e. improving standardization of iBCI systems evaluation and facilitating comparison between systems. The datasets used in the benchmark span a variety of tasks. A flexible evaluation interface is provided that allows easy submission of new iBCI systems. A range of methods are tested on the benchmark and results are reported and analyzed.

The assets as well as the evaluation platform could benefit future research on developing iBCI systems.

**Additional Feedback:**

This is an interesting work and could be valuable to the community. I have some concerns as mentioned above, but I am willing to reconsider my rating if they are addressed.

**Clarity:**

The paper is easy to understand for the most part, though I do find it hard to understand at times. This may partly be because some decisions and/or concepts lack description for someone not well versed in neuroscience literature. For instance, I did not know what "threshold crossing" was, and no explanation nor reference was provided where it was first mentioned (line 90).

Additionally, especially in the results section, description and reference is lacking sometimes. For instance, in section 4.2, a reference to Figure 4a is missing for the first paragraph, and it is not explained what the grey and colored lines are respectively in Figure 4b.

Finally, references to sources of data are missing in section 3.

**Correctness:**

Overall, the benchmark is constructed in a sound way, except for the concerns raised above.

**Documentation:**

The datasets and code are adequately documented and easily accessible. I want to commend the authors for the efforts in making them so.

**Ethics:**

I do not believe there is any ethical concern.

**Limitations:**

The discussion on limitations and potential social impact is adequate.

**Opportunities For Improvement:**

# Benchmark design

A large part of held-out sessions is redacted and is not used in either training or evaluation. I wonder why the redacted part cannot be used for evaluation?

The datasets contain neural data in the form of threshold crossings. Is that preferrable to raw neural data? If yes, why is that? If not, why is raw neural data not used?

Discussion on how these five datasets/tasks are chosen is needed, even if briefly. Are then chosen to be representative of iBCI decoding tasks that are of interest to researchers? Are they chosen based on data availability?

Definitions of the metrics are lacking. For instance, what is "variance-weighted average"? The choise of $R^2$ as the metric for movement tasks needs to be justified. How does it compare with other regression metrics, e.g. MSE? What are the caveats to bear in mind when interpreting it?

# Datasets

If I understand correctly, there is only one subject in each dataset? If so, I would expect the results obtained on the benchmark provide little hint as to how well a system would generalize to a different subject, and I think this is a main limitation. Is my concern valid? Or is that a known limitation of this work?

How is the length of sessions determined for each dataset, and how is the length of held-in and held-out sessions determined?

From line 145 to 146, the authors say "Human behavioral data are experimentally  instructed, while animal behavioral data are recorded from physical actions." What does "experimentally instructed" mean?

# Experiments

No model is used across all five datasets, if I understand correctly. Is that by design or is that a limitation imposed by other factors? In other words, are the tasks designed to be evaluated, and models compared, individually, or is it possible to compare models on the aggregated performance across tasks?

On line 256, it is said that the highest performing decoders from held-in sessions are chosen to evaluate on the held-out sessions. However, it is clear in Figure 3c-e that that is not the case. For instance, in Figure 3c, the RNN Oracle on day 3 and 4 perform better than on day 2, while day 2 is chosen as the RNN Static decoder.

The analysis of Figure 4b and 4c need to be elaborated. For instance, "The predictions support the R2 values shown in Figure 4a" (line 274) needs supporting arguments.

It is unclear over what the mean and standard variation is computed in Table 1. Are they over sessions or random seeds? And how does that translate to confidence intervals?

**Relation To Prior Work:**

The discussion on prior work is adequate.

**Summary And Contributions:**

This paper introduces FALCON, which is a benchmark for iBCI (intracortical brain-computer interfaces) systems. More specifically, FALCON consists of five datasets of neural and behaviour data collected from humans or monkeys in different tasks respectively, and FALCON provides a unified interface to evaluate on these tasks iBCI decoding systems that require varied levels of data availability.

The main contributions include:

- A standardized interface to evaluate and compare any iBCI system on five curated datasets.
- Experiment results and analysis of baselines and state-of-the-art methods on these datasets.
- Associated data and code.

---

> ### Author Rebuttal · Authors · 2024-08-15
>
> We appreciate the detailed feedback. We made many in-text changes in response to your comments and would be happy to share revised sections during discussion.
>
> ## Benchmark Design:
> - **“why the redacted part cannot be used?”**: Since iBCI decoding can be variable over days, it is important to include oracles to quantify good performance on held-out days. The redacted data was needed as training data for the oracles on held-out days and cannot be provided for evaluation. These oracles calibrate performance expectations for full calibration data vs the few-shot calibration data available publicly. We have added text to Section 2.1 to clarify.
> - **Why threshold crossings over raw?** Threshold crossings are used for most iBCIs as raw voltage recordings are much more noise-prone; threshold crossings filter for large deflections reflecting putative spiking activity from nearby neurons and have historical effectiveness for iBCIs. Ref 45 compares threshold crossings with other possible preprocessing in iBCIs. We've also clarified the text to define threshold crossings to non-neuroscientific readers.
> - **Task selection/motivation**: The datasets were curated for relevance to current iBCI research (studies in last 8 years). Datasets are individually motivated in their descriptions (Section 3) and on the FALCON website. Since we are exclusively re-using datasets originally collected for other studies, the datafiles selected were indeed constrained by availability. For example, H1 was curated from a series of experiments as the longest contiguous span of experiments with this behavioral task, and there is no further data available for this task that we are aware of.
> - **Metric definitions, justifications, limitations**: We have significantly revised the Metrics section of our manuscript to define and justify each metric choice more clearly. We also agree that noting caveats for interpreting each metric is an important addition, and have added a section to the Appendix noting the potential pitfalls of each metric.
> ## Datasets:
> - **One subject per dataset**: Indeed, collecting datasets for multiple subjects is a known difficulty in the field, particularly for our challenging task selection. Please see the common rebuttal for our response to this point.
> - **Length of sessions**: The total number of sessions were maximized from the set of available experimental data collected in a continuous span of time (e.g. several weeks). The held-in and held-out split was chosen by our discretion to balance the need for several sessions per split and the need for temporally contiguous splits. The number of trials used for few-shot calibration on held-out days was selected to ensure that new decoders cannot be trained to match performance of held-in calibrated decoders, as this would compromise the integrity of benchmarked results. These lengths are detailed in each dataset’s section of the Appendix.
> - **“Experimentally instructed”**: Human iBCI research participants typically have a severe condition resulting in complete or nearly complete paralysis, such as spinal cord injury or amyotrophic lateral sclerosis. In these human datasets, the researcher specifies desired behavior for the human to perform. For example, the human attempts or imagines movement according to visual cues. The behavioral data in the datafiles are these researcher instructions, synced to the neural data generated by the human during their behavior execution. These details are provided in the Appendix. We have also clarified this detail in Section 3 and eliminated the “experimentally instructed” phrasing.
> ## Experiments:
> - No model is used on all 5 datasets as a limitation of current models in the field. We chose representative high performance decoders in each of (M1, M2, H1), B1, and H2. While some stabilization procedures may be agnostic of behavioral task, their use with each type of decoder has not yet been demonstrated and may require significant experimentation. Developing new unified approaches for all datasets or modifying existing approaches to apply to new datasets was out of scope for our baselines. Hence, aggregate performance across tasks is also not a clearly defined metric at this stage of the FALCON benchmark.
> - **3c-e**: We agree that the figures may be confusing given the brief description, but we chose the decoders with the highest performance on other held-in sessions. The figures show per-session held-in decoder performance on their respective sessions; the metric for selection is not displayed, and may vary given the data quality in each session.
> - **4b/c**: We thank the reviewer for bringing to our attention that this section of the results is not clear. We have significantly revised the second paragraph of results Section 4.2 to walk readers through Figures 4b-c and their relationship to the summary of results in Figure 4a.
> - In Table 1, mean and standard deviation are indeed computed across sessions, not random seeds. We have updated the text to clarify. We would consider converting these to a confidence interval instead, but as these are metrics reported by the benchmark, such a presentation would be more confusing. Perhaps you could clarify if another formulation or presentation is preferred.
>
> ## Misc:
> - We now refer to Fig 4a in the first results paragraph.
> - We thank the reviewer for pointing out the missing details from the figure legend in Fig 4b. We now explain the gray and colored traces. Gray traces are the measured EMG, while colored traces are the predicted EMG for each method.
> - In each dataset’s description in Section 3, references to data sources are implicitly provided as citations of the works where experiments were originally performed. The Appendix provides additional delineated information including the links to each dataset on DANDI and more detailed acknowledgements of the data collectors and the original works.

---

> > ### Comment · Reviewer_pDQb · 2024-08-16
> > **Thanks for the response, some follow-up questions**
> >
> > Thank you for the response which helps answer many questions. There's still a few I'm not quiet clear or convinced about.
> >
> > **Why threshold crossings over raw?**
> >
> > I still think it'd be valuable to also release raw recordings even though threshold crossing is commonly used for iBCI. Moreover, the way thresholds are set seem arbitrary and some details are missing as well. E.g. "Thresholds to extract spiking data were manually set for each channel and may vary from session to session." (line 1084-1085 in supplementary material). Can you include more details of how it is done, and clarify how to ensure that the extracted data contains sufficient signal for the tasks?
> >
> > **No model is used across all five datasets**
> >
> > I agree that developing a unified method is out of the scope. However, for the argument that FALCON provides a flexible interface for evaluating new models, it would be great to demonstrate that one model can be used on all tasks even if it performs terribly. I wonder if this is worth having?
> >
> > **3c-e**
> >
> > I'm not sure I understand. Say for a task, there are in total $N$ held-in sessions $S_i$ where $i=1,2,\cdots,N$ is the session index, and on each session an oracle decoder $O_i$ is trained. Is the static decoder $O_s$ selected such that $O_s$ has the highest aggregate performance on all sessions except $S_s$?
> >
> > ---
> >
> > Additionally, can you share the snippets of changes related to the metrics as well as Figure 4b/c?
> >
> > Finally, I'd encourage you to add these responses to the paper:
> > - One subject per dataset: mention this as an limitation
> > - Table 1: mention that mean and std are computed across sessions, and that confidence intervals are not applicable since they are not IID samples

---

> > > ### Author Response · Authors · 2024-08-17
> > > **Response to follow-up**
> > >
> > > **Raw data**: We meant to mention (character limit) that raw recordings are typically at a 30kHz sampling rate per channel (as opposed to ~50Hz for typical binned spikes), causing practical burdens related to data storage and computational load for signal processing. Thus for many datasets, raw data were not stored/archived. And in the case of the clinical recordings, raw (broadband) data are not released due to participant privacy concerns (in case e.g. medications can be inferred from oscillatory activity, etc).
> > >
> > > However, we take the reviewer’s point that new stability approaches might be created if the community has access to broadband data. We do have access to the raw recordings for M2, and are happy to release those with an accepted manuscript. We are also actively looking into the availability of broadband data and possibility of its release for other datasets. This availability may lead to long-term innovation.
> > >
> > > **Threshold crossing processing details**: Experimenters unfortunately must often follow a semi-manual process during data collection. E.g., in H1, thresholds are initialized as a multiple of the RMS of recordings made in a baseline, pre-task period. Experimenters have discretion, varying across labs, to change these thresholds based on the noisiness of the filtered data; these details are documented in the primary sources for each dataset. Importantly, however, we know the threshold filtering is sufficient quality given the high performance of our held-in and oracle baseline models; additionally, each of these datasets were used for previous scientific studies with these thresholds.
> > >
> > > **No model on all five datasets**: We should clarify that while the FALCON few-shot abstraction is consistent across datasets, the tasks themselves are quite different. For example, the movement tasks require 50Hz continuous predictions of velocities or muscle activity, H2 is a trial-level classification task (discrete), and B1 is a trial-level high dimensional spectrogram prediction. The iBCI field currently lacks single-model unified methods (like a foundation model), but even the base architecture expected in any task can vary. Currently an RNN is applied to all but B1, but we are not aware of works that might apply an RNN to B1, and believe it out of scope to produce a new baseline not currently used in the field.
> > >
> > > **3c-e**:  This is correct. So to re-iterate, the plot shows the performance of each oracle on their own session, but the selection criteria is the performance on other held-in sessions.These are not the same due to day-to-day data quality variability.
> > >
> > > **Misc**: As mentioned in the shared response, we have added the one-subject discussion in the limitations and are also happy to share this text. We have also made the mean/stdev note in the table.
> > >
> > >
> > > ## Verbatim text requests
> > >
> > > **4b/c**: “”For Day 30, we also present decoded predictions for three key muscles - the biceps (BCPs), flexor carpi radialis (FCR), and extensor digitorum (EDC) - for each baseline approach. In \textbf{Fig. \ref{fig:m1_deep_dive}b}, each column is an individual reach for one of the location-object pairs available in the evaluation split on Day 30. Comparing the predicted EMG traces to the measured EMG traces provides context for interpreting the $R^2$ numbers and understanding which features of the EMG (the baseline, the high frequency features, the magnitude) are predicted well by each method. For example, NDT2 captures more high frequency changes in the muscle activations than other methods, potentially due to its nonlinear decoding. In \textbf{Fig. \ref{fig:m1_deep_dive}c}, we show $R^2$ values for each muscle individually. Per-muscle performance preserves the method ranking shown in \textbf{Fig. \ref{fig:m1_deep_dive}a}, providing further confidence that the variance-weighted $R^2$ over output dimensions is sound.””
> > >
> > > Metric verbatim text requests are posted in the following comment due to character limits.

---

> > ### Author Response · Authors · 2024-08-17
> > **Verbatim Metric requests**
> >
> > **Metrics**: We revised the metric introductions, and added a section on limitations to the Appendix.
> > _Introductions_: “Each dataset uses a standard decoding metric. The movement tasks (\mone, \mtwo, \hone) require predictions of multi-dimensional motor covariates, such as muscle activity. For these tasks, accuracy is reported using the coefficient of determination ($R^2$), computed as a variance-weighted average across the $R^2$ of individual motor covariates. $R^2$ is useful for interpreting low-dimensional predictions as a constant mean prediction achieves an $R^2$ of 0 and max $R^2$ is 1. The handwriting task (\htwo) evaluates predicted sentences, and thus we use word error rate (WER), computed as the edit distance between the predicted and expected sequence divided by the length of the intended sequence. Birdsong decoding (\bone) reports performance as mean squared error (MSE) on the predicted spectrogram; MSE is preferable for evaluating spectrogram predictions as the predictions are much higher-dimensional than in movement tasks. Metrics are computed per session and across-session mean and standard deviation are reported on EvalAI. Mean and standard deviation are computed separately for held-in and held-out splits.”
> >
> > _Limitations_: “One goal of the \falcon benchmark is to standardize the metrics used for evaluation of stable decoding performance for each task. As the decoded outputs are very distinct in nature, different metrics were selected for tasks in each domain, each one commonly used in their respective fields. As with all benchmarks, metrics should be interpreted with care, as these metrics alone do not necessarily capture all properties of a predicted output. $R^2$ has a convenient maximum at 1, but can be arbitrarily negative if the predictions contain more variance than the ground truth variable. $R^2$ also heavily penalizes predictions that are shifted from the expected center point. WER does not account for how close a given prediction is to the intended word and may overly penalize predictions that are only incorrect by a few characters. MSE can occupy unbounded ranges (i.e., [0, $\infty$)) that can be difficult to contextualize without other relative values. Hence, while models that demonstrate gains in the \falcon metrics will certainly show improved predictions, a poor-scoring model may not necessarily have unreasonable outputs. We recommend visualizing predictions to add additional context to \falcon scores.”

---

> > > ### Comment · Reviewer_pDQb · 2024-08-19
> > > **A few more follow-ups**
> > >
> > > Thanks for the response, I have two more follow-ups
> > >
> > > **Raw data and threshold crossing**
> > >
> > > Please mention that due to data availability the recordings are released as threshold crossings, and I'd encourage you to release M2's raw recordings.
> > >
> > > As for the methodology of setting threshold, I think it's a limitation of this work that thresholds are set arbitrarily and with varied methodologies, and this needs to be mentioned. I am not entirely sure how much signal is lost due to converting raw recordings to threshold crossing (e.g. how much better the performance would have been). But overall this is an acceptable limitation especially given that previous works also do this.
> > >
> > > **3c-e**
> > >
> > > Okay thanks for clarifying. What is the aggregated performance used for selecting the oracle exactly?
> > >
> > > I assume that the rationale is to pick the oracle based on generalization capability, but I suspect that this evaluation is not equally fair to all sessions. Assuming sessions that are closer in time have a smaller distribution shift, which I think is a reasonable assumption, one could argue that an oracle trained on a session with sessions that are close in time would be at an advantage compared to an oracle from a session that does not have any session that is close in time. E.g. if sessions are available on day 1, 5, 6, 7, the oracle from day 1 would be at a disadvantage compared to from day 5, 6, 7.
> > >
> > > This is a minor point to be fair, and I think it would be okay if text is modified in the paper to clarify how the static oracle is selected.

---

> > > > ### Author Response · Authors · 2024-08-20
> > > > **Respone to follow-up 2**
> > > >
> > > > **Raw data**: This request sounds reasonable. We will add to the limitations the following text: "FALCON uses exclusively spiking activity for decoding. While spiking activity is the default input for many BCIs, the experimental procedure for determining spiking thresholds can involve researcher discretion. Generally, thresholds are set as a multiple of the RMS of voltages recorded during a baseline period, but the precise multiple and protocol for baseline collection varies from dataset to dataset. To encourage research into stability methods that might avoid human variability or thresholding overall, we are additionally releasing the raw broadband activity for M2."
> > > >
> > > > **3c-e**: The aggregate performance used for oracle selection is the performance on the held-in block, excluding the session used to train the oracle, to evaluate generalization and avoid biasing for oracles trained on longer sessions. Your point on a possible bias is valid, though note that closer sessions do not always generalize better to each other (see [16], [17], [22] in main text). We will mention that this was one of several possible oracle selection schemes in the text.

---

> > > > > ### Comment · Reviewer_pDQb · 2024-08-20
> > > > > **One more follow-up**
> > > > >
> > > > > **3c-e**
> > > > >
> > > > > I think the oracle selection is okay. Just so I understand correctly, every oracle is evaluated on all other held-in sessions (excluding the one it is trained on), and the oracle with the highest average $R^2$ across these sessions is selected?
> > > > >
> > > > > Once this is resolved I'm ready to raise my rating.

---

> > > > > > ### Author Response · Authors · 2024-08-20
> > > > > > **Response to one more follow-up**
> > > > > >
> > > > > > Yes, that's correct. Thanks for engaging! Your feedback has helped catch several of our blind spots.

---

### Official Review · Reviewer_emfG · 2024-07-26
**FALCON Benchmark for Neural Decoding**

**Rating:** 5
**Confidence:** 4

**Review:**

Quality:
The FALCON benchmark is well-constructed, providing detailed descriptions of the datasets and the baseline algorithms tested. The inclusion of various datasets, such as those involving humans, monkeys, and a bird, enriches the scope and applicability of the benchmark. The choice of a standardized evaluation platform ensures consistent comparisons, enhancing the quality and reliability of the results.

Clarity:
The paper is generally clear in its presentation, with thorough descriptions of data collection processes and baseline models. However, there are some areas where clarity could be improved, such as specifying the reasons behind the selection of certain standards, like the NWB standard, and clarifying participant conditions. Some figures and results were not fully explained, which could hinder comprehension for readers not familiar with the domain.

Originality:
FALCON stands out for its focus on the standardization of evaluation metrics for neural decoding algorithms, particularly in the context of few-shot and zero-shot learning. This addresses a significant gap in the current landscape, where ad hoc comparisons often limit the progress and reliability of new methods. The inclusion of diverse datasets adds to the originality, offering a broad testing ground for various neural decoding approaches.

Significance:
The benchmark's significance lies in its potential to streamline and standardize the evaluation of iBCI decoders, a critical need in the development of practical, real-world BCI systems. By providing a consistent framework for comparison, FALCON facilitates the identification of robust algorithms that can handle the nonstationary nature of neural data, thus contributing to the advancement of BCI technology.

Pros:

Comprehensive Dataset Collection: Includes diverse datasets from multiple species and tasks.
Standardized Evaluation: Provides a unified platform for comparing different neural decoding algorithms.
Baseline Models: Offers initial benchmarks that facilitate understanding and further development.
Cons:

Clarity Issues: Some aspects, such as the choice of standards and dataset-specific details, could be better explained.
Limited Participant Diversity: Each dataset involves data from a single participant, which may limit generalizability.
Lack of Results for Some Models: Not all discussed models have their results presented, limiting a comprehensive view of their performance.
Overall, the FALCON benchmark is a valuable addition to the field of neural decoding, offering a structured approach to evaluating and improving iBCI systems. Addressing the noted clarity issues and expanding participant diversity in future datasets could further enhance its impact and utility.

**Strengths:**

The FALCON benchmark is a significant contribution to the neural decoding and brain-computer interface (BCI) communities. It offers a comprehensive and diverse set of datasets, including unique data from humans, monkeys, and a bird, which enhances the scope of research and testing. The benchmark's standardized evaluation platform ensures consistent and reliable comparisons of different neural decoding algorithms, addressing a critical need in the field. Additionally, the inclusion of baseline models provides a solid foundation for further research and development. The project is ethically sound, adhering to standards for data collection and use, and its emphasis on few-shot and zero-shot learning approaches is particularly relevant for developing robust, real-world BCI applications.

**Additional Feedback:**

Expand Dataset Diversity: Consider increasing the number of participants in each dataset to improve the generalizability of findings and robustness of the benchmark.
Detailed Benchmarking Results: Including results for all discussed baseline models would provide a more comprehensive understanding of their performance.
Ethical Transparency: Further detail on the ethical protocols followed, especially concerning animal data and human participant consent, would be beneficial.
Long-term Maintenance Plan: Outline a clear plan for dataset maintenance, including updates and access, to ensure ongoing relevance and usability.
Overall, the FALCON benchmark is a valuable resource for the research community, and addressing these suggestions could further enhance its impact and utility. Thank you for your contributions to this important area of research.

**Clarity:**

The paper is generally well-written, providing a clear overview of the FALCON benchmark, its datasets, and evaluation methods. The structure is logical, and the descriptions are detailed, making it accessible to readers familiar with the field. However, certain sections, particularly those explaining the choice of standards and specific participant conditions, could benefit from more detail. Additionally, ensuring all figures and results are clearly presented and explained would further enhance the paper's clarity. Overall, the paper effectively communicates its objectives and contributions, with minor areas for improvement in detail and explanation.

**Correctness:**

The claims made in the submission appear to be correct and are well-supported by the detailed descriptions of the datasets and baseline models. The datasets are constructed in a sound manner, with clear documentation of the data collection and processing methods. As a benchmark, FALCON provides appropriate and well-designed evaluation methods, ensuring consistent and fair comparisons of neural decoding algorithms. The use of a standardized evaluation platform adds to the robustness of the experimental design. However, providing more comprehensive results for all baseline models discussed would enhance the clarity and completeness of the evaluation. Overall, the submission is technically sound and meets the stated objectives.

**Documentation:**

The FALCON benchmark provides comprehensive documentation on data collection, including detailed descriptions of the experimental setups and data processing methods for each dataset. The submission includes baseline models and evaluation protocols, supporting reproducibility and consistent comparisons across studies.

**Ethics:**

The submission generally adheres to ethical guidelines, particularly concerning data collection and participant consent. However, there are areas where further discussion could be beneficial:

Research Involving Human and Animal Subjects: The use of data from humans and animals (monkeys and birds) should include detailed descriptions of ethical considerations, such as consent processes for human participants and welfare protocols for animals.
Data Privacy and Consent: Clarification on how participant data, especially for human subjects, is anonymized and protected is necessary to ensure privacy and compliance with legal standards.
Responsible Use: Providing guidelines on the ethical and responsible use of the datasets, including potential biases and limitations, would enhance transparency and ethical compliance.
Addressing these areas in detail would strengthen the ethical framework of the submission and align it with best practices in research ethics.

**Limitations:**

The authors acknowledge some limitations, such as the restricted number of participants in each dataset, which may affect the generalizability of the findings. However, there could be a more thorough discussion of potential negative societal impacts, such as the ethical considerations of using animal data and ensuring the privacy and consent of human participants.

Constructive Suggestions:

Expand Participant Diversity: Including data from more participants would enhance the robustness and applicability of the benchmark.
Ethical Transparency: A detailed exploration of ethical considerations, including data privacy, consent procedures, and the use of animal subjects, would strengthen the ethical framework of the research.
Long-term Maintenance Plan: Outlining a plan for the maintenance and updating of the datasets and benchmark would ensure ongoing relevance and utility.
Overall, the authors' openness about limitations is commendable, and addressing these suggestions would further solidify the benchmark's contributions to the field.

**Opportunities For Improvement:**

Clarity: The paper could improve clarity by providing more detailed explanations of certain choices, such as the selection of the NWB standard and the specific conditions of participants. Additionally, clearer presentation of all baseline results would help in comprehensively understanding model performances.
Participant Diversity: Expanding the datasets to include multiple participants for each task would enhance the generalizability of the findings and provide a broader basis for evaluating the robustness of neural decoding algorithms.
Ethical Considerations: While the paper adheres to ethical standards, a more detailed discussion on the ethical implications of using animal data and ensuring data privacy and consent, especially for human participants, would be beneficial.
Addressing these areas could further strengthen the benchmark's utility and impact in the research community.

**Relation To Prior Work:**

The paper effectively highlights the gap in standardization for evaluating neural decoding algorithms in brain-computer interfaces and positions the FALCON benchmark as a solution to this issue. It differentiates itself from previous contributions by emphasizing the use of few-shot and zero-shot learning approaches, and by providing a diverse set of datasets along with a standardized evaluation platform. However, the discussion could be further strengthened by explicitly comparing FALCON to existing benchmarks or datasets in the field, detailing specific advancements or unique aspects introduced by this work. This would help to better contextualize the significance of FALCON within the broader research landscape.

**Summary And Contributions:**

The paper introduces the FALCON benchmark suite, designed to standardize the evaluation of few-shot and zero-shot algorithms for neural decoding in intracortical brain-computer interfaces (iBCIs). The benchmark includes five datasets covering a range of motor and communication tasks, with data from humans, monkeys, and a bird. Each dataset includes baseline models to facilitate consistent comparison across methods. FALCON aims to enhance the robustness of iBCI decoders by providing a unified platform for testing and comparing algorithms, thus supporting the development of more reliable real-world applications.

---

> ### Author Rebuttal · Authors · 2024-08-15
>
> We thank reviewer emfG for taking the time to provide this review.
> ## Limitations
> **Limited Participant Diversity**: Please see the common response for details on this concern. We have added a section to the limitations in the text explaining the difficulty of increasing subject count.
>
> **Maintenance Plan**: The Systems Neural Engineering Lab (snel.ai) commits to maintaining the resources (website, codepack) provided in the FALCON project. Regarding future datasets: as new questions in the field emerge, we prefer that future benchmarks are used rather than expanding FALCON. We are certainly happy to provide our open-source infrastructure in support of such efforts.
>
> **Correctness**:
> The reviewer states that more comprehensive results for baseline models would aid the completeness of the paper. If you could provide detail on where more comprehensive results would be helpful, we could try to address that more directly. Importantly, not all baselines can be applied to every dataset, as the different iBCI scenarios entail different decoding strategies and model architectures (e.g., continuous decoding of velocities or muscle activity for movement tasks, versus discrete decoding for handwriting, versus sound synthesis for birdsong). This is also described in our response to the Experiments concerns of Reviewer pdQb.
>
> ## Clarity:
> **Use of NWB standard**: We chose the NWB data format as it is the predominant standard format for sharing neurophysiological data. Many of the datasets on the DANDI neurophysiological data archive use this format, and it has been used in previous ML-related neuroscience challenges such as the Neural Latents Benchmark (NeurIPS Datasets and Benchmarks, 2021).
>
> **Incomplete figures / results**: We are unclear on what the reviewer means by asking for clearer presentation of figures or baseline results. We have added text revisions to define neuroscience terms and to improve the clarity of the Results section in our response to other reviews (see pdQb). If you have additional concerns, could you please specify so that we might address them?
>
> **Relation to prior work**:
> FALCON is a benchmark of BCI decoding like the other benchmarks mentioned in the Related Work section. The challenges FALCON aims to evaluate are known problems, but the field lacks recent benchmarks with contemporary tasks and benchmarks, such that recent models mainly use custom evaluation. This motivation is described in detail in our Introduction. If there are specific works that you feel we are missing, we would appreciate a pointer to those.
>
> **Ethics**:
> We thank the reviewer for their attention to the ethics of our benchmark. First, we note that no new data was collected specifically for the FALCON Benchmark; all data were previously collected for other works that are explicitly described in each dataset’s section of the Supplemental Material. The full details on the experimental protocols for data collection, including prior ethical approvals and informed consent procedures, are available at the primary references for each dataset, which are linked in the supplemental descriptions and referenced to in the Ethical Considerations section of the text.
>
> We address further comments as follows:
> - Regarding the participant conditions: We were unsure what this comment referred to exactly, but interpreted it to refer to the participants’ etiologies of motor impairment. The primary references in the Supplemental Material provide detailed descriptions in that regard. We have clarified the text under Datasets to make it clear that human participants have limited independent movement capabilities. We would be happy to address any more specific questions you have.
> - Regarding the ethical implications of animal datasets: We have now added discussion of this topic in the Ethical Considerations section of the text. The text is reproduced below for convenience:
> > “FALCON also makes use of previously collected animal datasets. Animal models are critical to neuroscientific research that aids in improving our understanding of the brain and develops medical devices for the treatment or assistance of neurological disorders. We hope that by releasing standardized animal datasets, the FALCON benchmarking effort will contribute to the minimization of redundant data collection by allowing researchers to make better use of existing data.”
> - Regarding data anonymization: Human datasets are fully anonymized in that no personally identifying information is attached. All names are removed and data is released with participant informed consent as described in the primary sources of each dataset.
> Regarding responsible use: The FALCON benchmark releases only previously collected, fully anonymized data which poses no immediate risk to the participants as released. All datasets are hosted on DANDI under the CC-BY-4.0 guidelines for reuse.

---

### Official Review · Reviewer_dJra · 2024-07-29
**Invasive BCI benchmark across movements and communication tasks (5 datasets)**

**Rating:** 8
**Confidence:** 4
**Correctness:** I did not notice any flaw.
**Clarity:** very clear

**Review:**

Paper addresses a critical problem in iBCI literature that lacks standard datasets and benchmarks. This should enable
more solid and trustworthy method development in this field.

Papers mentions the 4 typical setups of iBCI: zero shot, few shot with labels (fine tuning), few shot without labels (domain adaptation),  test time adaptation (domain adaptation but with behavioral priors). Note that from an ML perspective the difference between few shot without labels and test time adaptation is subtle.

Paper is clearly written, comes with open data, open evaluation setup and good baseline code implementation on github. This work is a valuable contribution to the field of ML for iBCI.

**Strengths:**

Paper comes a solid literature review
Paper is well illustrated
Paper offers solid baselines with shallow and deep methods

**Additional Feedback:**

none

**Documentation:**

very good. All datasets come with getting started notebooks https://github.com/snel-repo/falcon-challenge/tree/main/data_demos

**Ethics:**

ok

**Limitations:**

An important limitation is that evaluation may be susceptible to promoting models that exploit the trial structure that is implicit in the datasets. This is however well acknowledged by the authors.

**Opportunities For Improvement:**

- Add refs in paragraph from L150 to 154 especially regarding the “anecdotal evidence“ claimed.
- on M1 task it is listed that 16 muscles are collected with iEMG. Please add details (potentially in Appendix) on the type of electrodes and how the muscles are targeted and identified.
- Please add a ref on the “Wiener Filter (WF; ridge regression with history) on inferred neural firing rates derived from an exponential spike smoothing kernel” and/or point to the appendix.
- Check acronyms are in capital letters in biblio eg ecog -> ECoG
- Some references have no journal or preprint link like 34 / 35 / 67.

**Relation To Prior Work:**

very good

**Summary And Contributions:**

The paper contributes a standard invasive BCI benchmark with 5 datasets across motor and communication tasks.
Data are obtained with microelectrode arrays or Utah arrays across humans or animals (birds, monkeys). Data are spikes.

A public Leaderboard is hosted on https://eval.ai while enforcing that the decoder operates on a stream of data while being causal.
This covers the open loop setting (no feedback to the subject).

Code: https://github.com/snel-repo/falcon-challenge/tree/main
Data available from https://dandiarchive.org/ eg https://dandiarchive.org/dandiset/000941 under CC-BY-4.0

---

> ### Author Rebuttal · Authors · 2024-08-15
>
> We thank the reviewer for their many specific suggestions and have integrated them into the manuscript as follows:
> - We have added citations to L150-154 to give readers more context about why we include both animal and human models.
> - We have rephrased the “anecdotal evidence” reference in L150-154 as we agree with the reviewer that this is unclear if evidence cannot be cited.
> - We have added details to the appendix to describe the EMG electrodes and placement procedures, including adding citations that detail the surgery.
> - We have added a reference to the appendix in the first paragraph of the Results section where we describe the Wiener Filter.
> - We have updated all references in the bibliography to correct any capitalization errors and to ensure all references have appropriate journal or preprint links.

---

### Official Review · Reviewer_zXpq · 2024-07-31
**FALCON benchmark project**

**Rating:** 8
**Confidence:** 4
**Correctness:** Yes.
**Clarity:** Very much.

**Review:**

Datasets and baselines are well chosen and represent a wide spectrum of iBCI problems. Baselines range from simple linear methods to more complex transformer architectures and provide a strong stating point for benchmarking work.
The datasets are indeed downloadable and results for baselines ran on EvalAI.

The overall problem is an excellent choice, and to highlight the applied impact, an initiative like this has the potential to actually benefit patients concretely by orienting methods researchers towards the problems that require attention in order to develop more effective and useful iBCI systems.
What cannot be measured, cannot be improved, and by providing FALCON this work paves the way for development of future methods that will perform better in the real world and reduce patient burden by potentially reducing the number of recalibrations sessions required.

**Strengths:**

- choice of problem that has potential for real-world impact
- methodology, composition of datasets, benchmarks, scientific documentation and insightful baseline analysesd
- clarity and quality of writing and visualizations

**Additional Feedback:**

This is an important contribution and the work has a high level of quality already. I hope the authors can benefit from the review to improve usability of the benchmark tools and examples.

**Documentation:**

Website and EvalAI and Github are generally in good shape, but some more testing and improving of instructions is needed to avoid the "works on my machine certificate".

**Ethics:**

No.

**Limitations:**

Yes.

**Opportunities For Improvement:**

Literature review
- literature review perhaps a bit one-sided, over-proportionally featuring iBCI work from certain research groups but not others
- literature review potentially a bit shallow on work for non-invasive EEG-based BCIs (e.g. see https://iopscience.iop.org/article/10.1088/1741-2552/aadea0/meta, https://arxiv.org/abs/2404.15319 for non-invasive EEG)

Code
- In a clean conda environment, I could install falcon but I had difficulties running the example code for the h1 dataset advertised on README, it fails complaining about 'local_data/sklearn_FalconTask.h1.pkl'
- I could not run the jupyter notebook for m1 either, it expects some missing local data (for loading the image) that do not come with the DANDI dataset for h1, also there are undefined variables preventing me from going beyond cell 2.
- general remark: not using data downloaders and letting users do renaming causes discomfort but perhaps also complexity
- I would expect some more work on testing / polishing to ensure people can rapidly test the examples and get started piloting their own models

Supplement / documentation
- Authors provide a notation for ridge regression that uses a diaognal regularization matrix, which suggests that different $\lambda$ values were used per feature but this is not the case when looking at the source code. This can create some confusion. Instead of $R^{\top}R$ authors could use the simpler $\lambda I$ expression or, alternatively, highlight the utility of using the more expressive regularizer in the context of expressing structured penalties in the Wiener Filter model, e.g., to apply more regularization to specific lags.
- Of note, authors could conveniently use RidgeCV() instead of a grid search using Ridge(), it could also speed up computation

**Relation To Prior Work:**

Yes in general, but see suggestions on literature.

**Summary And Contributions:**

The paper addresses the unmet need for benchmarking tools and dataset for neural data collected with intracortical brain-computer interfaces (iBCI), focusing in particular on the problems non-stationarities pose to neural decoders.
The paper presents a dataset challenge hosted on EvalAI with 5 cross-species datasets alongside a public library and example code for running benchmarks and - importantly – extensive scientific documentation of the data and problems.
Main results from the work well expose the main issue the benchmark aims to address by demonstrating the generalization failure of static or naive decoders (without specialized methods for tackling data drifts / use of recallibration). Moreover the analysis highlights the general difficulty of generalization when considering proper held-out datasets.

---

> ### Author Rebuttal · Authors · 2024-08-15
>
> **Literature review**: Thanks for this suggestion. We have integrated the MOABB reference given its interesting complement to our focus. We also agree that our bibliography may be overly focused on works from specific groups. Because the FALCON benchmark focuses on decoding from invasive recordings via spikes, our literature review of non-invasive decoding benchmarks serves only as an introduction to example related works. We have updated our introduction to include expanded citations from first and senior authors that were previously not represented.
>
> **Code “works on my machine”**: Thank you for trying out the release! We also agree the notebooks were not previously usable out of the box, thanks very much for pointing this out. We’ve implemented a few polishing steps to help reduce onboarding friction and have updated our documentation and notebooks accordingly. Notably, we have added a script that automatically downloads all datasets into the appropriate location, and all notebooks now provide an informative error if data has not been downloaded.
>
> **Ridge regression notes**: Thanks for catching the notation error. We have updated the manuscript accordingly (keeping the current formulation of applying the same regularization to all features). On the use of RidgeCV - our [codebase does currently use RidgeCV](https://github.com/snel-repo/falcon-challenge/blob/969663ef89a8098549b5e4595e09a0bb884eb186/decoder_demos/decoding_utils.py#L107) in the example if sweeping HPs. This may have been a change that came online after your review.

---

> > ### Comment · Reviewer_zXpq · 2024-08-20
> >
> > Thanks for your reply and the proposed action. Only 2 minor follow ups:
> >
> > "(1)  Thanks for catching the notation error. We have updated the manuscript accordingly (keeping the current formulation of applying the same regularization to all features) "
> >
> > In my understanding that would be a change to $I \lambda$ notation, which would implement a global regularizer. Are we on the same page?
> >
> >
> > "On the use of RidgeCV - our codebase does currently use RidgeCV in the example if sweeping HPs. This may have been a change that came online after your review."
> >
> > You use a grid search wrapper there, RidgeCV is a separate thing using efficient generalzied cross validation:
> >
> > https://scikit-learn.org/stable/modules/generated/sklearn.linear_model.RidgeCV.html

---

> > > ### Author Response · Authors · 2024-08-20
> > > **Follow-up**
> > >
> > > Thanks for engaging!
> > > 1. That's correct.
> > > 2. We'll take a look at the efficiency gains, thanks for the suggestion!

---

### Official Review · Reviewer_N3aA · 2024-08-01
**FALCON review**

**Rating:** 6
**Confidence:** 4

**Review:**

**Quality/Originality**

-	Diverse iBCI decoding tasks with data from human and non-humans.
-	Models can be submitted to the benchmark via EvalAI, a platform for evaluating machine learning models.
-	Supports a variety of learning strategies: zero-shot, few-shot supervised, few-shot unsupervised, test-time adaptation
-	Data files provided in an open format (NWB, Neurodata Without Borders), facilitating reusability
-	Data partitioned into calibration and held out sets
-	Data over several days to enable longitudinal analysis

**Clarity**

-	The paper is generally written well, and adequate documentation is provided for the platform, dataset and baseline analysis.

**Significance**

-	Provides a research tool for evaluating algorithms for iBCI long-term use.

**Strengths:**

Outlined above.

**Additional Feedback:**

See comments above.

**Clarity:**

-	The paper is generally written well. Figures and tables are well captioned and annotated.
-	Define all acronyms in the table.

**Correctness:**

Datasets are constructed in a sound way (open file format provided via a public repository). Evaluation methods and experiment design for benchmark (documentation) are appropriate and performed well.

**Documentation:**

- Benchmark platform and evaluation code are hosted on a public website and a URL is provided.
- Methods are detailed in the main paper, supplement or referenced to the relevant publications.
- Some unpublished models are included.
- A summary of the datasets is included in the paper and related publications are referenced. Dataset documentation on the public repository needs to be improved.

**Ethics:**

-	Ethical concerns arise due to research involving animal, humans and the invasive nature of sensors (intracortical). Authors acknowledge ethical concerns around iBCI technology.

-	Previously published datasets are used in the framework. Authors indicate that approval was obtained prior to conducting animal (Institutional Animal Care and Use Committees) and human (Institutional Review Board) studies from respective ethical boards. Human trials were conducted under FDA investigational device exemption (iBCI).  For human studies, authors note that informed consent was obtained prior to experiment procedures.

**Limitations:**

- Authors acknowledge limitations of the platform. FALCON evaluates only open loop predictions.

- Issue of small sample size, how platform will accommodate other datasets.

- See other comments above.

**Opportunities For Improvement:**

- Small sample size in referenced datasets (only one subject per dataset).

- It is not clear if/how the inclusion of other datasets/open file formats will be accommodated on the platform (this could address the limited sample size issue). Note that additional datasets will need to have undergone ethical review.

- While contact information is provided, it is not clear how the website will be maintained long term.

- More documentation is needed on the dataset repository (DANDI). Dataset, file structure, etc. should be described to stand alone without necessarily referencing the FALCON website. See FAIR data principles.

**Relation To Prior Work:**

-	Authors discuss their work in relation to previous contributions in the literature.  There are limited datasets/benchmarks available for iBCIs.

**Summary And Contributions:**

This paper presents FALCON, an open platform to evaluate adaptation algorithms for intracortical brain-computer interface (iBCIs). Datasets are hosted on DANDI, a public repository; includes neural and behavioural data from decoding tasks in humans (handwriting, robotic effector) and animals (monkey reach and grasp, monkey finger movement, songbird vocals) over several days. Utility of the FALCON platform is demonstrated with several baseline models.

---

> ### Author Rebuttal · Authors · 2024-08-15
>
> **Increased subject sample size**: See the common rebuttal for our comment on increasing sample size.
>
> **DANDI Accommodations**: Regarding the reviewer’s concern about dataset revisability on DANDI -- should more datafiles be added, DANDI allows for straightforward editing of the dataset page to add additional files, revise existing files, or modify dataset descriptions accordingly.
>
> **Website Maintenance**: The Systems Neural Engineering Lab (snel.ai) commits to maintaining the FALCON benchmark and associated assets.
>
> **Dataset Documentation**: We thank the reviewer for highlighting the need for standalone dataset documentation and have updated the DANDI descriptions on each dataset’s page accordingly.
>
> **Table 1 Acronyms**: Regarding the reviewer’s request to define all acronyms in Table 1, many of these acronyms are defined in the table’s legend. Some acronyms (e.g., Wiener Filter (WF), Recurrent Neural Network (RNN), and Language Model (LM)) are defined throughout the text. The reviewer makes a good point that other methods are also acronyms that are not necessarily defined elsewhere. This is our oversight, and we have updated the text to make sure they are included. These are:
> - Neural Data Transformer 2 (NDT2)
> - Cycle-consistent Generative Adversarial Network (CycleGAN)
> - Nonlinear Manifold Alignment with Dynamics (NoMAD)
> - Continual Online Recalibration with Pseudo-labels (CORP)
>
> **Ethics**: We cannot tell whether the reviewer has a concern about the ethics of our benchmark specifically or is simply flagging the sensitivity of the topic. We agree with the reviewer that iBCI technology is a sensitive topic. All data was collected under and approved in prior ethics reviews. For this release, the human data is anonymized. We do not foresee risks specific to the study of stable decoding in this framework.

---

> > ### Comment · Reviewer_N3aA · 2024-08-26
> >
> > Reviewer appreciates the authors' responses to the issues raised. A couple of follow-up concerns/questions:
> >
> > - Subject sample size: Reviewer understands the challenges with conducting studies with invasive BCIs. However, the authors are proposing iBCI benchmarks. Even with subject specific models, there needs to be a few subjects for consideration or else the entire benchmark performance evaluation will be biased to only one subject. From the authors’ main response, they indicate they reached out to “multiple iBCI groups” and there was potential to generate datasets with multiple subjects. The justification for focusing on more challenging tasks is not strong. Could the authors elaborate on what these “simplistic tasks” are? A multi-subject iBCI data would still have high value (note that this is not about generic models but subject-specific models).
> >
> >       "There is a tradeoff here: we could have instead chosen datasets that focused on more simplistic tasks, where multiple subjects may have been available."
> >
> > - Dataset updates: If additional subjects are going to be added to the dataset (M1 dataset?), then this needs to be noted in the description and the dataset version needs to be indicated. This is relevant when reporting/comparing results with a significant change in the dataset.
> >
> > - Dataset documentation: Authors added more documentation to the DANDI website. Improvements/clarifications are still needed.
> >
> >      o- Complete all the metadata elements. For example, the H2 dataset includes the ethics approval (clinicaltrials.gov, FDA IDE and IRB protocol numbers for human studies; or IACUC for animal studies), but this information is not available in the other datasets. (H1 dataset has this information in the description but not in dataset metadata)
> >
> >      o- The “Number Of Subjects” is listed as 3 across all the datasets.
> >
> > - The EvalAI link appears to be broken based on this error: *“Challenge does not exist!”*

---

> > > ### Author Response · Authors · 2024-08-28
> > > **Response to Comment**
> > >
> > > **Subject sample size**:  FALCON reflects the field in its current data constraints -- to clarify, we did not collect (generate) additional experimental data for this work. In our selected tasks, there are simply no additional available datasets or subjects to be provided for any of the FALCON datasets except for M1. High quality multi-subject datasets for iBCI are quite rare, and further requiring a contiguous block of sessions for evaluating decoder stability is an even stricter criteria, typically requiring months to years of effort. Such effort usually results in a separate publication and dataset release. For example, one potential multi-subject dataset we considered, from [17], is already released for its associated paper; this release and others do not withhold high quality data for private evaluation, disqualifying it from use in a benchmark.  We do suggest that submitting to multiple tasks (which intrinsically spans subjects) is a natural path for mitigating the single-subject bias.
> > >
> > > We use the term 'simplistic tasks' for behavioral tasks with a small number of behavioral conditions; a canonical example is 8-direction center-out reaching. These datasets are limited for benchmarking stability for two primary reasons. First, the behavior is quite simple and not reflective of desired real-world iBCI applications. Second, the behavior is highly structured, which tends to be much easier to align across sessions compared to the tasks selected for FALCON. For example, in such structured tasks, one could develop a trivial solution that essentially memorizes and reproduces such structured patterns across days. It is therefore critical to use tasks with more behavioral complexity.
> > >
> > > **Dataset updates**: We agree. We will make sure to do this when we add the additional subject to the M1.
> > >
> > > **Dataset Documentation**: Thank you. We have added the ethical protocols to the  metadata for H1, B1, M1, M2. Associated papers are already provided in the metadata. We have not added associated projects as FALCON does not yet have a DOI, and subject matter is provided in the description. The number of subjects is automatically generated by DANDI according to the number of folders provided, and is unfortunately not editable. We have added a statement to the dataset description to make it clear that each folder is a dataset split, and there is only data from one subject in each dataset at present.
> > >
> > > **EvalAI Link**: Thanks for pointing out this error - the link is live now.

---

### Author Rebuttal · Authors · 2024-08-15

We thank the reviewers for their time and feedback. We’re pleased that all reviewers found the problem area important and the work timely. The reviewers also found the writing and figures clear, and we will incorporate specific suggestions provided into the final revision, as noted in individual responses. Similarly, the reviewers found the additional resources around the challenge (documentation, codebase) generally clear, and we will integrate feedback as the benchmark opens up. Finally, on the benchmark itself, the reviewers appreciated the diversity of datasets and tasks evaluated in FALCON (N3aA, zXpq, emfG, pDQb), and the baselines provided (zXpq, dJra).

A common concern (N3aA, emfG, pDQb) was the use of a single subject or participant in each of the 5 datasets. We completely agree with the reviewers that, in the long-term, evaluating decoding stability across several subjects per task would be an extremely useful benchmark. However, there is a more pressing problem: all modern high-performance iBCI demonstrations to-date have used single-subject modeling, and few among these have demonstrated robustness to temporal nonstationarities. Thus while there is great enthusiasm for multi-subject models, we focused our benchmark on the most immediate problem for the BCI community: robustness of single-subject models.

In this scope, we have selected tasks that are relatively complex and at the frontiers of real-world BCI, and to our knowledge FALCON’s datasets are unique in their breadth of behaviors and provided temporal spans. To achieve this complexity and diversity, we brought together multiple iBCI groups, each at the forefront of their respective problems. There is a tradeoff here: we could have instead chosen datasets that focused on more simplistic tasks, where multiple subjects may have been available. However, in order to introduce the larger community to challenging problems that will push the capacities of modern BCIs, we necessarily had to choose scenarios where data from only a few subjects are available. Of the datasets released through FALCON, only the M1 dataset (monkey reach and grasp) has another high-quality, multi-session dataset from an additional subject available. We commit to releasing the data for this subject through the FALCON benchmark prior to the camera-ready deadline, if accepted, in an effort to begin working towards multi-subject datasets with relevance to BCIs.

As dataset quality can vary widely across subjects, we believe claims about subject generalization would require a cohort of more than two or three subjects -- and it is currently prohibitively expensive to collect datasets with such high numbers of subjects. This underlying subject generalization concern is an important one, and we note that the framework established in this FALCON benchmark can easily be applied to future works once the data for such an effort is collected. We have added discussion of this limitation to our manuscript.

---

### Decision · Program_Chairs · 2024-09-26

**Decision:**

Accept (Poster)

**Comment:**

This submission generate interest and discussion. The reviewers which engaged in the discussion found that the benchmark was solid, with many baselines and a good literature review, and could help with real-world impact.